# Understanding how Victoria, Australia gained control of its second COVID-19 wave

James M. Trauer [1✉], Michael J. Lydeamore [2,3], Gregory W. Dalton[3], David Pilcher[1], Michael T. Meehan[4], Emma S. McBryde [4], Allen C. Cheng[1,3], Brett Sutton[3] & Romain Ragonnet [1]

During 2020, Victoria was the Australian state hardest hit by COVID-19, but was successful in controlling its second wave through aggressive policy interventions. We calibrated a detailed compartmental model of Victoria's second wave to multiple geographically-structured epidemic time-series indicators. We achieved a good fit overall and for individual health services through a combination of time-varying processes, including case detection, population mobility, school closures, physical distancing and face covering usage. Estimates of the risk of death in those aged ≥75 and of hospitalisation were higher than international estimates, reflecting concentration of cases in high-risk settings. We estimated significant effects for each of the calibrated time-varying processes, with estimates for the individual-level effect of physical distancing of 37.4% (95%CrI 7.2−56.4%) and of face coverings of 45.9% (95%CrI 32.9−55.6%). That the multi-faceted interventions led to the dramatic reversal in the epidemic trajectory is supported by our results, with face coverings likely particularly important.

[1] School of Public Health and Preventive Medicine, Monash University, Melbourne, VIC, Australia. [2] Central Clinical School, Monash University, Melbourne, VIC, Australia. [3] Victorian Department of Health, Government of Victoria, Melbourne, VIC, Australia. [4] Australian Institute of Tropical Health and Medicine, James Cook University, Townsville, QLD, Australia. ✉email: james.trauer@monash.edu

The COVID-19 pandemic has had an unprecedented impact on human health and society[1,2], with high-income, urban and temperate areas often the most severely affected[3–5]. The impacts of the virus are felt directly through its substantial infection-related mortality[6,7] and post-infection sequelae[8], as well as through the often highly restrictive public health measures needed to achieve control[9].

Australia was relatively successful in controlling COVID-19 throughout 2020[10], with all jurisdictions of the country achieving good control of the first wave of imported cases through March and April. However, the southern state of Victoria suffered a substantial second wave of locally-transmitted cases, reaching around 600 notifications per day, predominantly in Metropolitan Melbourne in winter.

In response to the pandemic, the Victorian Government implemented a number of recommendations and policy changes with the aim of reversing the escalating case numbers that had a severe impact on social and economic activities. Specific changes included stringent restrictions on movement, increased testing rates, school closures, and face-covering requirements. In Metropolitan Melbourne face coverings were mandated from 23rd July and significantly more stringent movement restrictions were implemented from 9th July (moving to "stage 3") and from 2nd August (moving from "stage 3" to "stage 4"). Stage 4 restrictions included business closures, restriction of restaurants and cafes to take-away service only, remote schooling, restriction of travel to a five-kilometre radius, an 8 pm curfew, and reduction in public transportation[11], constituting one of the world's longest and toughest lockdowns[12]. Case numbers peaked in the final days of July and the first days of August and declined thereafter.

Understanding the relative contribution of each of these interventions is complicated by several interventions being implemented within a few weeks, along with policy differences between metropolitan and regional areas. Although several countries of Asia also maintained effective control strategies through much of 2020[13], Victoria's second wave was notable in that these policy changes reversed substantial and escalating community cases rates and supported subsequent sustained elimination, which was achieved for several months from November 2020. The clear reversal in the trajectory of the epidemic following the implementation of these policy changes offers the opportunity to explore the contribution of these factors to the epidemic profile. We adapted our computational model to create a unified transmission model for the state and infer the contribution of the policy interventions implemented to changing the direction of the epidemic trajectory.

## Results

**Calibration fit**. We achieved good calibration fits to all calibration targets (Fig. 1 and Supplemental Figs. 7, 8), along with close matches to health service cluster-specific (henceforward "service") indicators not used for calibration (Supplemental Figs. 9–11), under the framework of a single state-wide model. The modelled epidemic peaks in the regional services occurred somewhat later than in the metropolitan services, which is attributable to the modelled infection first being seeded in the metropolitan regions before triggering epidemics outside of Greater Melbourne and is consistent with historical reality. These fits were associated with a post-wave proportion of the population recovered of around 1%, with higher proportions in metropolitan regions and young adults (Supplemental Fig. 20).

**Parameter estimation**. The posterior estimates of model calibration parameters are presented in Table 1 and posterior histograms of key parameters of interest in Fig. 2. Several

epidemiological parameters with good evidence from international studies showed posteriors that were consistent with prior beliefs. This prevented overfitting, reduced the degree of freedom, and provided better estimates of key free parameters including the effect of time-varying processes, allowing insights into the dynamics of the epidemic. The unadjusted risk of transmission per contact (specifically the risk of transmission per contact between a susceptible person aged 15–64 years and a symptomatic infectious person not in isolation in any location) was estimated at 4–6%. This needed to be adjusted for contiguous groups of services, with the modifiers applied to the metropolitan services reaching values up to double that for the regional services (other than Barwon South West). Consistent with our intuition around these parameters, the location-specific adjusters to the contact rate were generally correlated with one another, but anti-correlated with the transmission risk parameter. The extent of mixing between neighbouring geographical patches was low, with around 0.6–4.7% of the local force of infection contributed by regions neighbouring the index patch.

Estimates of the incubation period, the infectious period, and the period prior to ICU admission were similar to our prior estimates derived from the literature. However, the risk of hospitalisation, and hence of ICU admission among those infected, was considerably greater than our age-specific prior estimate obtained from the literature. Similarly, the risk of death in those aged 75 and above was considerably higher than that typically reported in the literature. This likely reflects higher rates of exposure and infection in population groups at particularly high risk of adverse outcomes, including residents of aged care facilities.

The case detection rate associated with a testing rate of one test per 1,000 population per day was not markedly constrained by fitting to data and was estimated at 17.9% (95%CrI, 8.1–28.5), such that peak rates of detection of symptomatic infections were estimated at greater than 60%.

To understand the reasons behind the epidemic curve peaking at the start of August and beginning to decline thereafter, we were particularly interested in parameters governing the effect of time-varying processes. We estimated that physical distancing behaviours and face coverings were both important in achieving control of Victoria's second wave, with face coverings estimated to have reduced transmission and infection risk by around 33 to 56%. The effect of physical distancing behaviour was less constrained through calibration, but was estimated to have reduced the risk of transmission/infection by 7 to 56%. The smaller changes in reported adherence to this intervention (Supplemental Fig. 5) meant that this had a lesser impact on the epidemic profile. For the two behavioural changes, the posterior probability density was substantially more informative than the prior and had lower density around the value of zero, consistent with an effect of each of these interventions in reversing the epidemic trajectory. Additionally, the posterior probabilities of the parameters were only moderately collinear (Supplemental Fig. 14), supporting independent effects for each process.

Initiating our calibration algorithm from a diverse set of starting points did not influence our parameter posterior estimates, supporting the ability of our model to define the high posterior regions of the multi-dimensional parameter space (Supplementary Section 14).

**Sensitivity analyses**. Using base contact survey data from Belgium instead of those from the United Kingdom resulted in negligible differences to our analysis. Similarly, when residential Google mobility was used to scale home contacts instead of

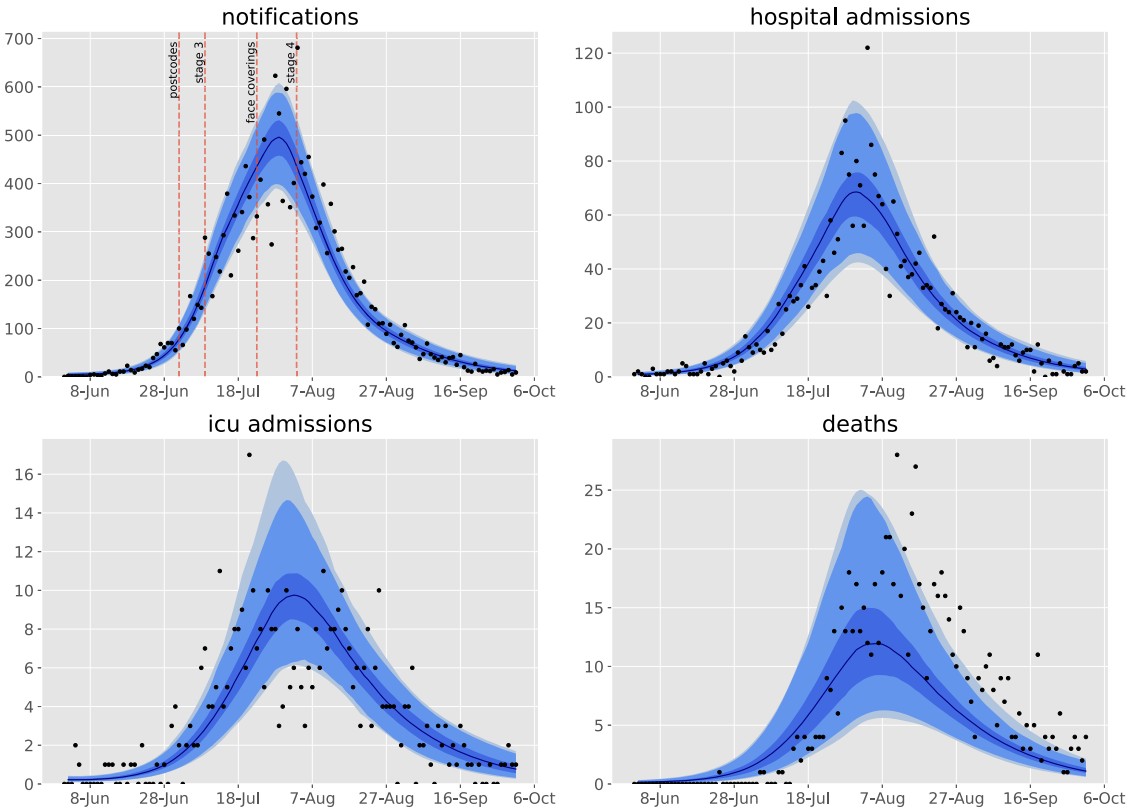

**Fig. 1 Calibration fits to daily state-wide time series of notifications, hospital admissions, intensive care unit admissions, and deaths.** Daily confirmed cases (black dots) overlaid on the median modelled detected cases (dark blue line), with shaded areas representing the 25th−75th centile (mid blue), 2.5th−97.5th centile (light blue), and 1st to 99th centile (faintest blue) of estimated detected cases. The timing of restrictions applied to Metropolitan Melbourne is indicated in the upper left panel.

keeping these contact rates fixed, the estimated posteriors for the behavioural processes changed moderately, with a greater estimate for the effect of physical distancing. This observation resulted from the need to scale back contacts in the three locations outside of the home more dramatically if the effect of home contacts increased during lockdown.

**Counterfactual scenarios.** Figures 3–5 present counterfactual scenarios compared to the baseline scenario. The effect of re-opening schools from 9th July (the date that stage 3 restrictions were imposed) was projected to be modest, with daily case rates peaking around 200 higher than under baseline conditions, but with the epidemic profile otherwise broadly similar. The effect of not mandating face coverings was projected to be dramatic, with case numbers in the thousands for several months under the counterfactual of face coverings usage remaining at the baseline level of 13.0%. Returning to full mobility from 9th July resulted in a similarly poorly controlled epidemic, under the assumption that face coverings usage could not then have reached the baseline estimate of >90% compliance in all workplaces and other locations if industries such as hospitality were fully re-opened. An epidemic unmitigated by any movement and behavioural restrictions was projected to substantially overwhelm expanded ICU capacity.

## Discussion
We found that the improvement in Victoria's second wave of COVID-19 cases could be well captured in our transmission model through a combination of time-varying processes that included: testing rates, population mobility, use of face coverings,

and physical distancing. The lower rates of COVID-19 observed in regional services were captured with the introduction of the infectious seed through the metropolitan services along with modest and plausible changes to the risk of transmission by geographical region. The risk of infection in metropolitan areas was estimated to be up to double that of regional areas, consistent with international findings of a moderate correlation between population density and epidemic severity[3]. Although Barwon South West showed transmission rates that were more comparable to Metropolitan Melbourne, this region includes Victoria's second-largest city of Geelong. Interaction between populations of different services was low in the context of significant restrictions on movement between regions. Each of the time-varying processes modelled appeared to be important to the observed dynamics, with both face coverings and behavioural changes associated with a significant reduction in transmission risk per contact. However, face coverings had a considerably greater effect on reversing the epidemic, which was observable due to the sharp transition in the extent of their use when they were mandated.

Victoria's second wave of cases was dramatically different from its first autumn wave, which was driven by importations and during which time the effective reproduction number was consistently estimated to be below one[14,15]. Victoria's second wave was initiated by quarantine escape, from which widespread community transmission soon followed. Progressively more extensive lockdown measures were then implemented, with local targeting of specific residential blocks and then postcodes, which were insufficient to reverse the epidemic trajectory.

As noted previously, stage 3 restrictions were associated with a reduction in the effective reproduction number[16], although

**Table 1 Prior distributions and posterior estimates with 95% credible intervals of all calibrated epidemiological model parameters.**

| Parameter (units) | Prior distribution | 2.5th centile | Median | 97.5th centile |
|---|---|---|---|---|
| Unadjusted risk of transmission per contact | Uniform, support: [0.04, 0.07] | 0.0413 | 0.0463 | 0.0602 |
| Incubation period (days) | Truncated normal, mean: 5.5, standard deviation: 0.97 support: [1, infinity) | 4.35 | 6.06 | 7.62 |
| Duration of active disease (days) | Truncated normal, mean: 6.5, standard deviation: 0.77 support: [4, infinity) | 5.23 | 6.47 | 7.86 |
| Pre-ICU period (days) | Truncated normal, mean: 12.7, standard deviation: 4 support: [4, infinity) | 6.72 | 13.0 | 20.0 |
| Infection fatality rate for ≥75 age group | Uniform, support: [0.05, 0.3] | 0.071 | 0.189 | 0.289 |
| Hospitalisation rate adjuster | Uniform, support: [0.5, 5] | 1.27 | 3.02 | 4.73 |
| Infectiousness of asymptomatic persons multiplier | Uniform, support: [0.2, 0.8] | 0.227 | 0.509 | 0.762 |
| Starting infectious population (persons) | Uniform, support: [20, 70] | 21.6 | 37.3 | 66.6 |
| Case detection rate at one test per 1,000 per day (proportion) | Uniform, support: [0.05, 0.3] | 0.081 | 0.179 | 0.285 |
| Proportion of contacts identified at prevalence of one case per 1,000 population | Uniform, support: [0.2, 0.5] | 0.209 | 0.341 | 0.487 |
| Inter-service mixing (%) | Uniform, support: [0.005, 0.05] | 0.00591 | 0.0217 | 0.0471 |
| Effect of physical distancing | Uniform, support: [0, 0.6] | 0.072 | 0.374 | 0.564 |
| Effect of face coverings | Uniform, support: [0, 0.6] | 0.329 | 0.459 | 0.556 |
| Reduction in the effect of home contacts | Uniform, support: [0, 0.4] | 0.0147 | 0.206 | 0.389 |
| Service-specific contact rate multipliers | | | | |
| North Metro and West Metro | Truncated normal, | 0.96 | 1.25 | 1.63 |
| South Metro and South East Metro | mean: 1, | 0.70 | 1.01 | 1.34 |
| Barwon South West | standard deviation: 0.5, | 0.58 | 0.95 | 1.41 |
| Other regional services | support: [0.5, infinity) | 0.53 | 0.71 | 1.01 |

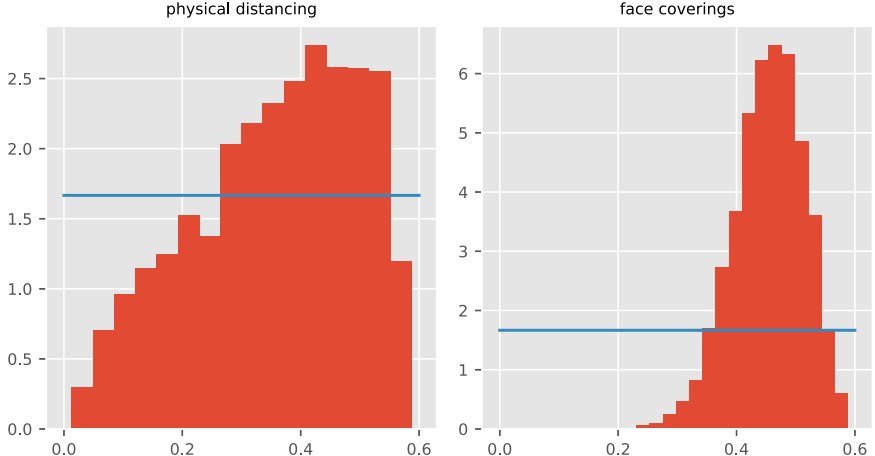

**Fig. 2 Posterior density histograms for key state-wide epidemiological parameters from accepted model runs.** Red histograms, model posterior estimates; blue lines, prior distributions for the same parameters (both uniform distributions).

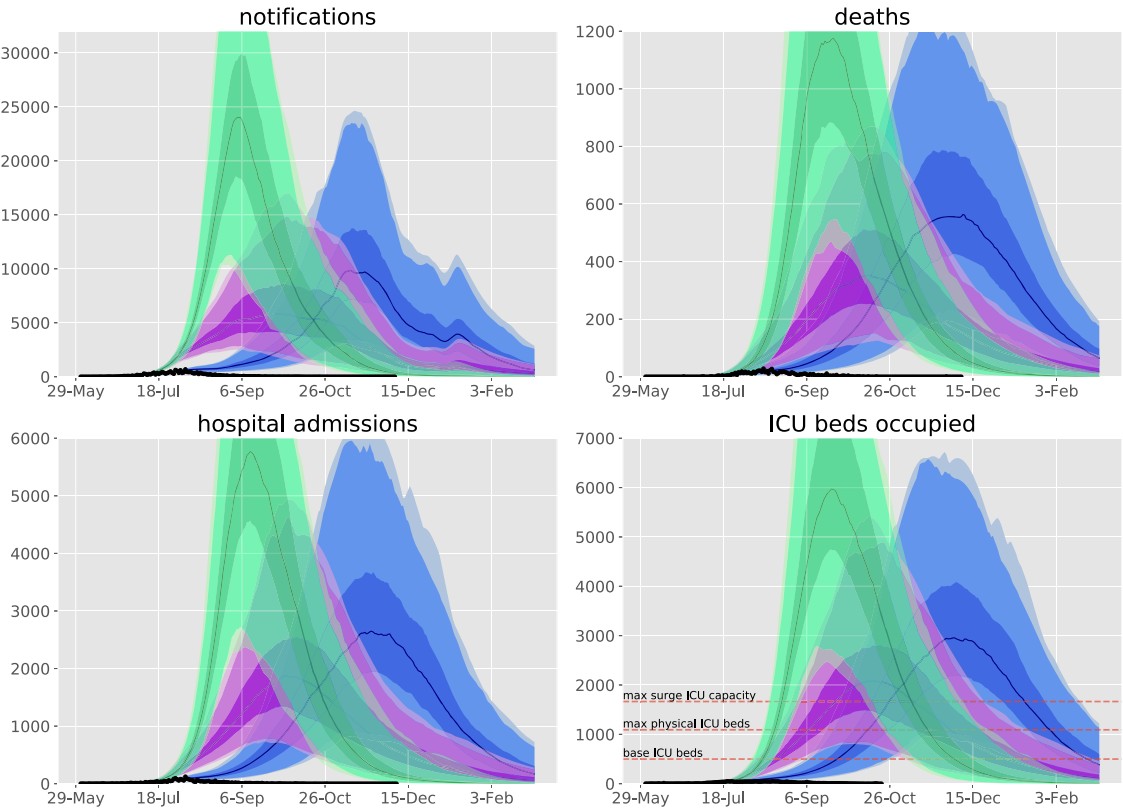

**Fig. 3 Counterfactual scenarios compared against baseline calibration and data.** Scenarios are: blue, face coverings not mandated with compliance remaining at base level of 13.0%; purple, work, education and other locations mobility return to baseline levels from 9th July with 60% face coverings compliance; green, work, education, and other locations mobility return to baseline levels from 9th July with face coverings compliance remaining at base level. Data (black dots), median modelled estimates (lines), shaded areas 25th−75th centile (darkest shading), 2.5th−97.5th centile (intermediate shading depth), and 1st to 99th centile (faintest shading) of each indicator for each scenario. 9th July chosen as the date that stage 3 restrictions were imposed. We considered that full compliance with mandatory face coverings would be impractical if workplaces and other locations returned to full capacity (for example, if hospitality was fully re-opened, patrons would not wear masks in all other locations). Base and surge ICU capacity for Victoria presented on lower left panel[38].

significant case rates persisted throughout July, and further reductions in mobility were observed with stage 4. An agent-based model with detailed social networks, consideration of multiple intervention types, and without geographical structure was calibrated to the Victorian epidemic[17]. This model emphasised the importance of associations between individuals who would not otherwise be in regular contact to the epidemic. Another agent-based simulation found that earlier activation of social distancing interventions could halve the total epidemic size[18]. By contrast to previous work, our model captures both the temporal and spatial implementation of the policy changes in Victoria to allow inference of the effect of each intervention. As concern increased that epidemic control had not been achieved over the course of July, the policy changed rapidly in an attempt to bring the epidemic under control. Testing numbers increased following a nadir in early June and lockdown measures were implemented differently in twelve Melbourne postcodes, the remaining postcodes of Greater Melbourne, Mitchell Shire (immediately north of Greater Melbourne) and the remainder of regional Victoria. We captured these complicated geographical patterns of restriction by scaling our mixing matrices using Google mobility data, which are available at the LGA level for Victoria. School closure and face-covering policy changes were captured according to the dates of policy changes.

In earlier versions of the model we included an effect of seasonal forcing. While good fits were also achieved with this effect included, the posterior estimate of the effect of seasonality was

not markedly constrained through fitting to data. The minimal information provided on seasonal forcing was likely attributable to our simulation period spanning less than four months and so covering a small proportion of the cycling period. Therefore, while a potentially important seasonal effect would be consistent with our analysis and with evidence from elsewhere[19], it was not possible to draw conclusions as to its strength. The effect of face coverings was similar to or greater than is typically estimated at the individual level[20,21], but is consistent with the dominant importance of the respiratory route to transmission[22]. The finding was also not unexpected given the marked shift in population use of face coverings at this time and the timing of the policy change in late July relative to the dramatic reversal in case numbers occurring around one week later. The significant estimated effect of behavioural changes suggests that reductions in interpersonal associations (macro-distancing) alone were not solely responsible for the dramatic reversal in the epidemic trajectory observed. However, the Google mobility functions used to capture macro-distancing simulated falls in attendance at workplaces and other non-household locations to considerably below baseline values in several services (Fig. 6), emphasising their importance. The dramatic effect of each of these interventions on the epidemic trajectory (relative to the parameter estimates that suggest relatively modest individual-level efficacy) is partly attributable to our implementation of these processes as applying to both the infectious cases and the exposed individual. This approach is analogous to simulating the use of bed-nets for

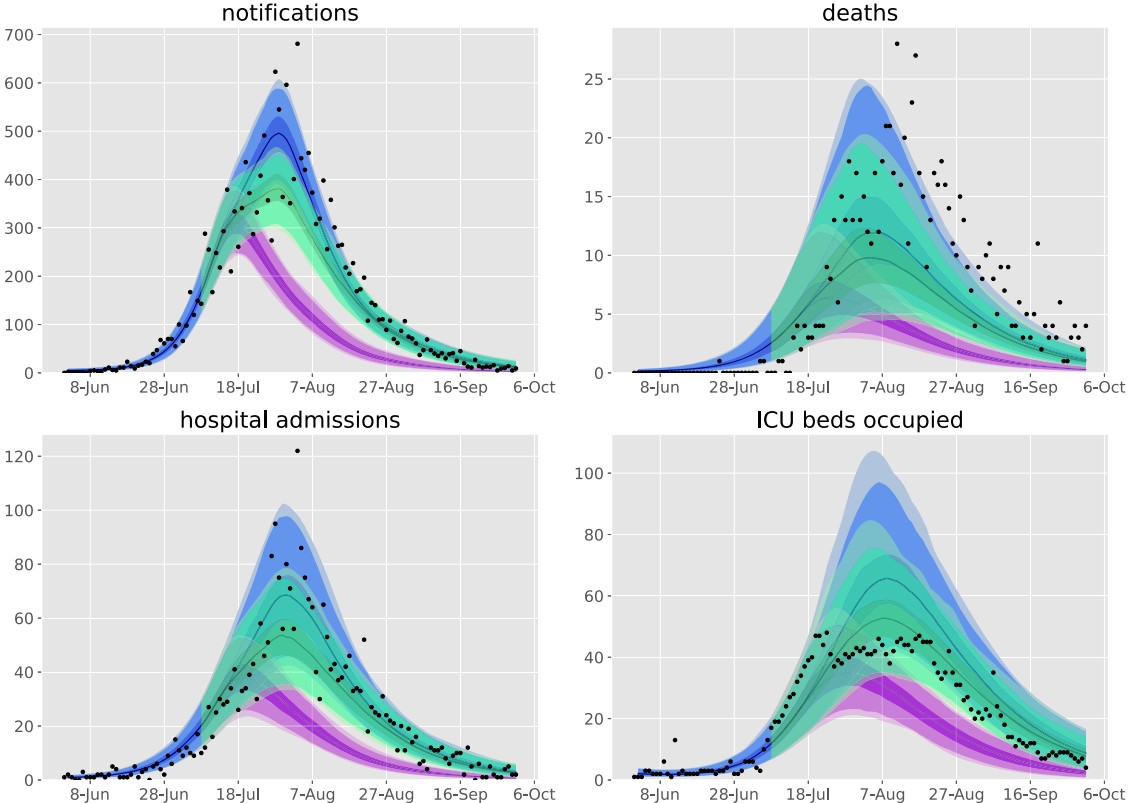

**Fig. 4 Counterfactual scenarios compared against baseline calibration and data.** Scenarios are: purple, face coverings policy introduced two weeks earlier with stage 3; green, stage 4 mobility levels (introduced 2nd August) instead commence from 9th July. Data (black dots), median modelled estimates (lines), shaded areas 25th−75th centile (darkest shading), 2.5th−97.5th centile (intermediate shading depth), and 1st−99th centile (faintest shading) of each indicator for each scenario.

malaria control, where the overall effect of the intervention is quadratic (i.e., the scaled transmission rate is effectively the square of the complement of the intervention effect), as it affects both the infection vector and recipient[23].

Compared to many other countries, Australia experienced a relatively minor COVID-19 epidemic throughout 2020 and relatively few serosurveys have been undertaken in the country. A national serosurvey undertaken in elective surgery patients in four states including Victoria in June/July 2020 identified no seropositive patients from 3,037 samples tested[24], supporting our approach of commencing our simulations with a fully susceptible population. A serosurvey of health care workers from Eastern Health in Melbourne's eastern suburbs found a seropositive proportion of 2.2% in November/December 2020 in a group at higher risk of exposure than the general community[25]. This is consistent with our estimates of a population-wide recovered proportion of around 1%, with marked differences by age group and location.

Despite the complexity of our model, it is inevitably a simplification of reality. The uncertainty regarding the effect of asymptomatic persons on epidemic dynamics was addressed by varying the relative infectiousness of these patients, which was not well-constrained through fitting, but suggested around twofold lower infectiousness per unit time. Our findings relating to the impact of time-varying interventions could be proxies for other behavioural changes, although our simulation has the advantage of having all time-varying processes informed through empiric data. With limited data for the profile of face-covering use in regional Victoria, we assumed this followed the policy change dates with an analogous shape to that of Metropolitan Melbourne.

In earlier analyses and manual exploration of parameter variation in our model, we found that fixing the importance of home contacts to be equal to that of those in the three other locations (i.e., omitting the home contact reduction parameter) required either a greater effect of face coverings or a shorter incubation period to achieve the observed downslope in case numbers in August and September. This alternative configuration increased effective home contacts relative to those in other locations during the epidemic peak and led to either implausibly high effects for face covering or a very short incubation period and hence serial interval. We also note that the use of face coverings may partially be a surrogate for other individual-level behavioural changes that are not captured through the survey responses that scale the physical distancing function in our model.

Victoria's second wave is known to have had particularly dramatic effects on residents of aged care facilities and health care workers[26], which we did not explicitly capture except by varying parameters relating to disease severity. The concentration of cases in aged care was likely the main factor requiring us to inflate the international estimate for the infection fatality rate for those aged 75 years and over. Our results suggest a markedly higher IFR in this group than that estimated from other settings, but is consistent with the high raw case-fatality rate of 4.3% in the data used for fitting (801 deaths, 18,459 notifications). This highlights the importance of risk factors and comorbid conditions on the estimated IFR, which likely underpin some of the dramatic increases in IFR with increasing age and are particularly concentrated in residents of aged care facilities. For these reasons, we emphasise that our forward projections (Figs. 3 and 5) of a lesser public health response assumed that the IFR for the oldest age group returned to the uninflated international estimate. Our inflation of the age-specific estimates of the risk of hospitalisation given symptomatic COVID-19 are also consistent with a more severe

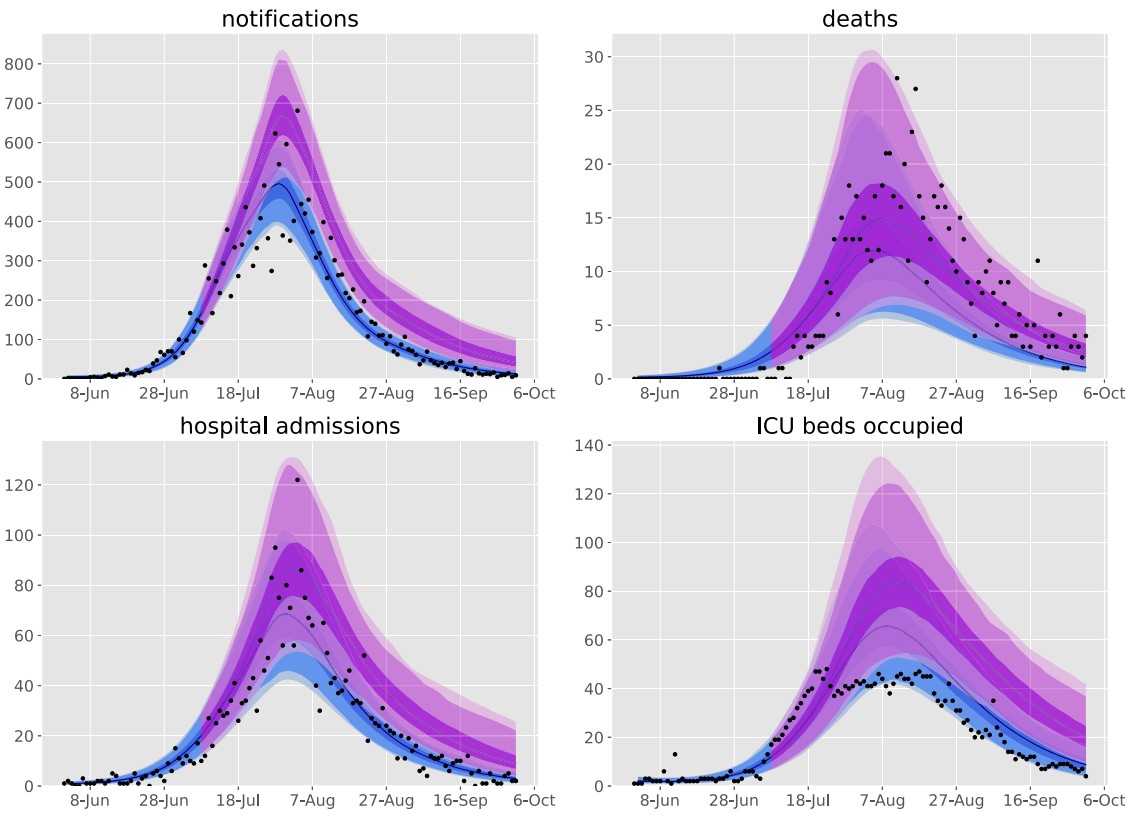

**Fig. 5 Counterfactual scenario of education remaining on-site, compared against baseline calibration and data.** Scenarios are: purple, school opening scenario; blue, baseline calibration. Data (black dots), median modelled estimates (lines), shaded areas 25th to 75th centile (darkest shading), 2.5th to 97.5th centile (intermediate shading depth) and 1st to 99th centile (faintest shading) of each indicator for each scenario.

epidemic, although hospital admission is driven by factors other than disease severity. These include infection control and work-force capacity with staff isolation requirements in residential aged care facilities, which were particularly important to this epidemic wave.

With the state's explicit objective of achieving no community transmission in Victoria (and therefore across Australia) within a few months[27,28], our findings emphasise that multiple interacting components of the public health interventions were required to achieve this within the modelled period[29,30], Consistent with findings from elsewhere[31–33], without reductions in contacts outside the home and mandating the use of masks, there would have been no reasonable prospect of driving transmission to zero within a time period tolerable to the community, given the starting point of the epidemiological situation in late July. Since the period modelled, Victoria has achieved several periods of sustained COVID-19 elimination, likely allowing restrictions to be released to an extent that would have been impossible otherwise[34]. The small effect of school closures was also consistent with findings from overseas[30,35], although if schools had remained open throughout the epidemic wave, some additional weeks would likely have been required for transmission to decline to the point that elimination was an immediate prospect. None-theless, it is encouraging that in a low transmission scenario prior to the emergence of variants of concern, school closures are likely not necessary to gain control in the presence of other effective population-level restrictions, including masks.

In conclusion, we found that Victoria's major second wave of COVID-19 was brought under control through a combination of policy interventions that were synergistic and together contributed substantially to the dramatic reversal in the observed epidemic tra-jectory. In particular, the considerable individual-level effect of face

coverings was critical to achieving epidemic control, and so should be a cornerstone of any public health response given the much lesser inconvenience associated with their use compared to restrictions on mobility. Rates of hospitalisation and death were higher than anticipated given international estimates of parameters pertaining to these quantities, likely reflecting the concentration of the epidemic in high-risk groups, particularly residents of aged care facilities. As vaccination is rolled out as a more targeted and definitive inter-vention to gain control of the pandemic, procurement, logistics and population confidence continue to limit the rate at which population immunity can be achieved, while variants of concern increasingly threaten control. In this context, understanding how public health and social measures can be efficiently deployed to regain temporary control while vaccination is deployed remains critically important.

## Methods

We adapted the transmission dynamic model that was previously used to produce policy-relevant analyses and projections of specific public health and social measures in 2020. The model was also used to forecast new cases, health system capacity requirements, and deaths for the Victorian Department of Health and Human Services (DHHS) and the Government of Victoria at the health service cluster level (hence-forward "service") during the second wave to December 2020. By incorporating geographical structure to represent services, we built a unified model of the COVID-19 epidemic in Victoria, and fitted the model to multiple indicators of epidemic burden in order to infer the effectiveness of each component of the response to the epidemic. Full methods are provided in the Supplementary Methods, key features of the model are illustrated in Fig. 1 and the version of the code used is tagged at https://github.com/monash-emu/AuTuMN/releases/tag/vic_2nd_revision.

**Base model**. Our model of COVID-19 epidemiology is a stratified, deterministic SEIR framework, with sequential compartments representing non-infectious and infectious incubation periods and early and late active disease (Fig. 6A) coded in Python version 3.6. The late incubation compartment and the two active com-partments are stratified to simulate epidemiological considerations including asymptomatic cases[36], incomplete detection of symptomatic cases, hospitalisation,

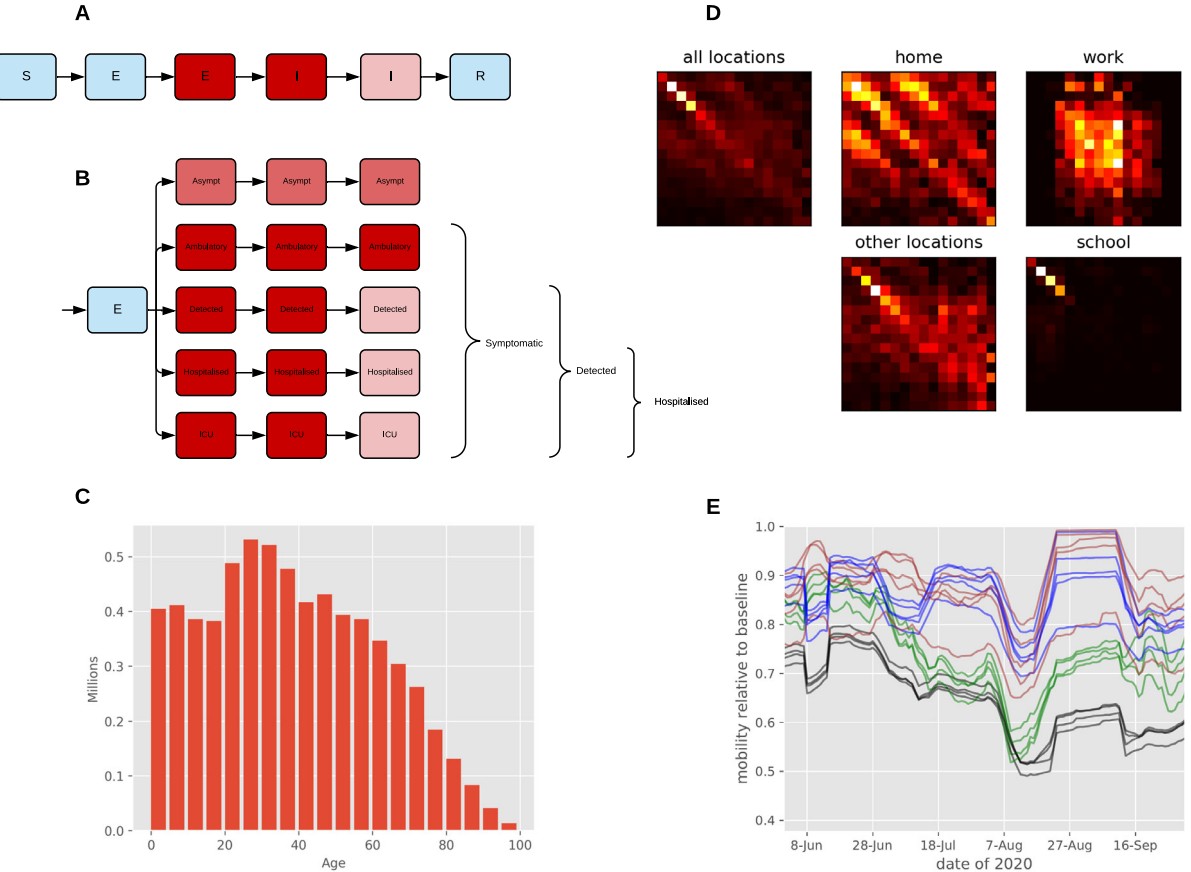

**Fig. 6 Age-structured COVID-19 model with population distribution, age-specific contact rates, and mobility inputs. A** Unstratified model structure (with example infectiousness shading for infected compartments). **B** Illustration of the "clinical" stratification used to capture infection, detection, and hospitalisation status, with a depth of red shading illustrating the infectiousness for infectious compartments (with light blue being non-infectious). Note that some or all compartments are further stratified by age, contact tracing status and health service cluster. **C** Starting population age distribution in five year bands starting from 0 to 4. All bands aged 75 and above were aggregated into a single modelled age group representing those aged 75 and above. **D** Heterogeneous mixing matrices by 16 age groups in the absence of non-pharmaceutical interventions. The intensity of yellow/red shading represents the number of contacts per day. **E** Macro-distancing adjustments to the mixing matrices for each service smoothed with 7-day moving average. Black, workplace mobility for metropolitan services; green, other locations mobility for metropolitan services; blue, workplace mobility for regional services; brown, other locations mobility for regional services. The horizontal axis is date of year 2020. The vertical axis indicates Google mobility estimate relative to the baseline pre-pandemic period in early 2020. S susceptible; E exposed; I active; R recovered; ICU intensive care unit.

and ICU admission (Fig. 6B). All model compartments were then stratified by age, with susceptibility, the clinical fraction, hospitalisation risk, and infection fatality rate modified by age group[6]. We introduced heterogeneous mixing by age using mixing matrices that we constructed by weighting the empiric age-specific contact rates for the United Kingdom from the POLYMOD study to the age structure of the Victorian population (Fig. 6D and Supplementary Section 1).

**Simulation of public health interventions.** We simulated movement restrictions (including school closures, business closures, and working from home) by varying the relative contribution of three of four locations to the overall mixing matrix (Fig. 6E) continuously over time. Using Google mobility data (https://www.google.com/covid19/mobility/) weighted to service, we scaled the work contribution with workplace mobility and the contribution from other locations (contacts outside of schools, homes, and work) with an average of mobility from the remaining Google mobility locations other than residential (Fig. 6E). We simulated school closures by scaling the school contribution according to the proportion of children attending schools on site. We assumed that schools began transitioning to onsite learning from the 26th of May, at which time 400,000 of 1,018,000 students returned to onsite education. The remaining students were considered to return onsite from the 9th of June, before 90% of students moved to remote learning from the 9th of July, which continued until October. We reduced the contribution of contacts in the home with a parameter that was constant over time to avoid over-emphasising the importance of this location. This approach was chosen to acknowledge processes that cannot be captured through a compartmental model, including that home-based contacts are subject to contact saturation and are less important to linking transmission chains together.

The term "micro-distancing" is used to refer to behavioural changes that reduce the risk of transmission given an interpersonal contact and so are not captured through data on population mobility (e.g., maintaining physical distance and use of face coverings). Micro-distancing was assumed to reduce the risk of both transmission from index cases and the risk of infection of susceptible persons, with the effect of both physical distancing and face coverings applied to all three non-residential locations. Both the coverage and the effectiveness of each intervention were incorporated, with time-varying functions representing the proportion of the population complying with recommendations over time and constant calibration parameters scaling these functions to represent the effectiveness of the intervention. The profiles of compliance with these two recommendations was estimated by fitting to YouGov data, available at https://github.com/YouGov-Data/covid-19-tracker, with hyperbolic tan functions providing a good fit to data (Supplemental Figs. 5 and 6). Because face coverings were mandated ten days later in regional Victoria than Metropolitan Melbourne, the face coverings compliance function was delayed by this period for regional services, while the physical distancing function was identical for all services.

We defined the modelled case detection rate as the proportion of all symptomatic cases that were detected through passive presentation to health care (Fig. 6B). We related the case detection rate (CDR, Eq. 1) to the number of tests performed using an exponential function, under the assumption that a certain per capita daily testing rate is associated with a specific case detection rate, with this relationship varied during calibration:

$$CDR(time) = 1 - e^{-shape \times tests(time)} \qquad (1)$$

To capture contact tracing, all model compartments representing current infection were stratified into traced and untraced strata, with infectiousness of all compartments of the traced stratum reduced to the same level as those admitted to hospital. The infectious seed was assigned to the untraced stratum and all new infections enter the incubation compartment through the untraced stratum of the early incubation compartment. The process of contact tracing is then represented as moving people from the untraced to the traced stratum of the early incubation period. The proportion of all persons moving to the traced stratum rather than continuing to progress through compartments in the untraced stratum is calculated as the product of the proportion of all active cases detected and a proportion representing the effectiveness of contact tracing ($q$(time)). The latter proportion is considered to decline as prevalence of infection increases according to Eq. 2. The parameter governing the relationship between infection prevalence and the effectiveness of contact tracing ($\tau$) is varied through calibration on the assumption that a certain infection prevalence is associated with a specific contact tracing effectiveness.

$$q(time) = e^{-prev(time) \times \tau} \qquad (2)$$

**Incorporation of health service clusters**. We further stratified the above model to Victoria's nine health service clusters, including four services which together constitute Metropolitan Melbourne (North, West, South, South East Metro) and five regional services which together constitute the rest of Victoria (Barwon South West, Gippsland, Grampians, Hume, Loddon-Mallee). We split the estimated age-specific population for Victoria (Fig. 6C) according to historical patterns of accessing health service clusters provided by DHHS. The infectious seed was split across the compartments representing current infection and assigned evenly across the metropolitan services, with the remainder of the population assigned to the susceptible compartments. The force of infection in each service was calculated using a spatial mixing matrix constructed based on the geographical adjacency of the clusters. The final model included 2,592 compartments interacting through a dynamic mixing matrix of dimensions 144 × 144 (16 age groups and nine geographical patches), with each matrix element scaling over time to reflect changes to population mixing in response to changes in mobility and pandemic-related policy decisions as introduced above.

**Calibration**. Because of the high-dimensional parameter space, we calibrated the model to reproduce local COVID-19 dynamics during Victoria's second wave using an adaptive Metropolis algorithm, which is non-Markovian but retains ergodic properties[37]. For the prior distributions of epidemiological calibration parameters, we used uniform priors for highly uncertain quantities and truncated normal distributions for quantities informed by epidemiological evidence (Table 1). To reduce the number of parameters estimated, we included "adjuster" parameters in our calibration to modify all of the age-specific proportions of symptomatic individuals together, and all of the age-specific proportions of symptomatic individuals who are hospitalised together. These adjusters are multiplicative factors that are applied to the odds ratio equivalent to the proportion parameter, rather than directly to the parameter value itself; thus ensuring that the adjusted value lies between zero and one. We also reduced the number of free parameters scaling the transmission rate in each of the services, using two scaling parameters across the four metropolitan clusters and two across the regional clusters.

The likelihood function was constructed by first incorporating Poisson distributions with rate parameters equal to the modelled daily state-wide estimates for each of notifications, hospitalisations, ICU admissions, and deaths. This was then multiplied by terms for the daily time-series of notifications for each service, smoothed with a four-day moving average, using normal distributions. As there is no requirement for individuals living in a service's catchment to attend that health service, we allocated each service a proportion of each notification according to the historical tendency of persons from each Local Government Area (LGA) to attend a hospital from that service (such that daily service-specific notification and death counts are not integer-valued). To demonstrate the ability of our calibration algorithm to define the highest posterior regions of our parameter space, we used Latin hypercube sampling to initialise parameter sets from diverse starting points across their prior distributions (Supplementary Section 14).

**Sensitivity analyses**. We ran two sensitivity analyses to vary key assumptions employed in the base analysis described above. First, we replaced the matrices derived from data from the United Kingdom with matrices similarly derived from Belgian data, given similar population demographics of this country. Second, we considered the effect of assuming that "home" location contact rates scaled with residential Google mobility estimates because this data source is qualitatively different from other Google mobility domains.

**Scenarios**. We considered counterfactual scenario projections based on the baseline calibration that included both earlier and later applications of the same interventions as were applied in reality. For projections considering lesser restrictions (and so a larger epidemic) we returned the infection fatality rate in those aged 75 and above to the baseline estimate (a weighted average of the infection fatality rate in those aged 75−79 and those aged 80 and above). We chose this assumption because the higher mortality rates observed during the second wave were likely attributable to higher rates of transmission in elderly persons with higher rates of comorbid illness. Therefore, we wished to avoid assuming that this observation would continue if widespread community transmission had occurred.

**Data ethics**. Notification, death, and health service capacity data (disaggregated by health service and date only) for calibration were provided to Monash and stored in encrypted form. Google's publicly available mobility data consist of aggregated, anonymised sets of data from users who have chosen to turn on the location history setting. The YouGov data used for constructing microdistancing functions are publicly available at https://github.com/YouGov-Data/covid-19-tracker.

**Reporting summary**. Further information on research design is available in the Nature Research Reporting Summary linked to this article.

## Data availability

All data other than that used for calibration are available at git tag https://github.com/monash-emu/AuTuMN/releases/tag/vic_2nd_revision and are deposited with Zenodo at https://zenodo.org/record/5553691#.YV7OSdrZU2w. The input data (including mobility estimates) used in this analysis are contained in the SQLite database at AuTuMN/data/inputs/inputs.db of the tagged repository. The output data from our models are available at http://www.autumn-data.com/app/covid_19/region/victoria/run/1627427875-54bf2e3.html.

## Code availability

All code is available at the git tag and Zenodo repository listed above. The current version of the repository for infectious disease model construction is at https://github.com/monash-emu/summer, with documentation at http://summerepi.com/index.html.

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

## Acknowledgements

We gratefully acknowledge the support and advice of staff of the Victorian Department of Health and Human Services (now the Victorian Department of Health) for the provision of data and assistance with its interpretation. We thank Prof Nicholas Golding for providing the micro-distancing compliance functions.

## Author contributions

J.M.T. and R.R. constructed and analysed the model. M.J.L. and G.W.D. assisted with the provision of data. D.P. advised on the implications for ICU capacity. A.C.C. and B.S. advised on the implications for pandemic management. M.T.M., R.R. and E.S.M. checked the mathematical approach and expressions. J.M.T. wrote the first draft of the manuscript, which was then revised with input from all authors.

## Competing interests

B.S. and A.C.C. wish to emphasise their important statutory roles during Victoria's pandemic response in 2020, as Chief Health Office and acting Chief Health Officer respectively. M.J.L. and G.W.D. were also employed by DHHS during 2020. JMT provided regular advice to DHHS during this time as an independent advisor. The Epidemiological Modelling Unit of Monash University provided the health system cluster-level projections for notifications, admissions, and deaths under contract to the Victorian Department of Health and Human Services in 2020. JMT is a recipient of an Early Career Fellowship from the Australian National Health and Medical Research Council (APP1142638). The other authors declare no competing interests.
