## [Peer Review File · Nature Communications]

REVIEWER COMMENTS

Reviewer #1 (Remarks to the Author):

This study used a detailed age- and location-stratified compartmental model to explore the COVID-19 outbreak in Victoria, Australia, from June through October 2020. The key findings relate to the calibration of the model to the epidemic data via an MCMC algorithm, which allows the determination of parameter values, which are then interpreted in terms of their broader implications (e.g., the relatively high efficacy of face masks). The study draws on a wide variety of different data sources, including epidemiological data, mobility data, demographic data, etc. The findings are broadly consistent with previously published estimates, e.g. regarding face mask efficacy and other parameters.

MAJOR COMMENTS

This study has much to commend it, including the use of a carefully designed, open-source modeling library and the use of a large number of different data sources. However, I had a number of concerns regarding the methodology. Most seriously, almost all of the paper's results hinge on the calibration. A large number of parameters were simultaneously calibrated (20), despite the fact that relatively little data was available to calibrate to. While there were different regions, most regions had similar temporal trends, and experienced the same interventions at the same times (with some exceptions regarding regional vs. metro areas), so these regions are not really independent.

Put simply, the data comprise a noisy set of data points that comprise a near-symmetric curve. It is somewhat unclear to me how one could constrain reasonably well even 3-4 parameters with these data, much less 20 parameters (not to mention that these parameters were also combined with a time-varying 144x144 mixing matrix, which has its own considerable uncertainties associated with it). Not enough details are given about the calibration algorithm to independently validate it. Not only do the parameter distributions look curiously Gaussian, but there is much less degeneracy or cross-correlation between the parameters than one would expect. For example, couldn't one get identical results by multiplying each of the cluster-specific contact rates by a factor of 2 and then dividing the unadjusted risk of transmission by a factor of 2? Trade-offs such as these should both smear out the univariate parameter distributions and introduce (anti)correlations between them; the fact that these do not appear makes me think that parameter space was not explored sufficiently by the MCMC algorithm, rather than that the distributions truly are (almost) all well-constrained Gaussians.

There are several other methodological concerns, described in more detail below. In my view, most serious of these are (a) the way the case-detection rate was calculated (which does not seem to be correct in a context of low case numbers and active contact tracing); (b) the large correction factor required for the infection-fatality ratio, which to me suggests some methodological concerns; and (c) the use of Google mobility data, which does not seem to be consistent with other local mobility sources for Melbourne (e.g. Apple, Facebook), nor is it consistent with international sources (e.g. the claim that Melbourne, under strict lockdown, experienced a smaller drop in mobility than some US cities that had only weak mobility restrictions).

Finally, after crafting such a rich model, I was surprised that the authors placed so much emphasis on the calibration step, which is often -- if unfairly! -- regarded as a nuisance rather than "real" results. The scenarios section was intriguing, but was quite brief, and I felt many stones were left unturned in terms of (a) exploring how many infections and deaths could have been averted by an earlier face mask requirement, (b) comparing differences between regions (clusters), (c) exploring whether fits of

similar log-likelihood could have been achieved with a much simpler model (e.g., apply the Bayesian information criterion), etc. The paper is already of reasonable length (especially with the supplement), and I am not suggesting that the authors make it even longer. However, I do wonder whether a more impactful paper could be written by taking empirical best-estimate parameters -- for both face mask efficacy and infection-fatality rates, for example -- at face value, instead of trying to calibrate them, and then use this as a more robust starting point to explore quantities that are truly unknown, such as the case detection rate at different points in the epidemic, or even the change in transmission due to spatiotemporal changes in mobility.

MINOR COMMENTS

- Line 21: "The impacts of the virus [...]" -- I found this sentence a little confusing.
- Line 33: It would be helpful to explain to readers what each of these stages indicate; a reader unfamiliar with the Victorian context would not appreciate that "stages 3 and 4" translate to Melbourne experiencing "one of the world's longest and toughest lockdowns" (<https://www.bbc.com/news/world-australia-54654646>)
- Line 40: "the experience of Victoria's second wave is virtually unique in that the pattern of substantial and escalating daily community cases was reversed following these policy changes, with elimination subsequently achieved in November" -- It's not clear to me what aspect of this is being flagged as unique; much of this is similar to the experiences of other countries that achieved (temporary) elimination, such as New Zealand, South Korea, Vietnam, etc.
- Line 48: It might help to explain what a "cluster" means here, since it is often used in contrast to widespread community transmission (e.g., the Cedar Meats cluster), rather than as a component of it.
- Line 62: Is 2-5% per contact per day? Is this the same for households, workplaces, and the community?
- Line 65: What data were used to constrain the geographical mixing? I did not see this described in the supplement.
- Line 70: "However, the risk of hospitalisation (and hence ICU admission) and of death given infection were considerably greater than our age-specific prior estimates obtained from the literature. This likely reflects higher rates of exposure and infection in population groups at particularly high risk of adverse outcomes, including residents of aged care facilities." -- While the cause might be correct (possibly worth citing [https://www.thelancet.com/journals/lancet/article/PIIS0140-6736\(20\)32206-6/fulltext](https://www.thelancet.com/journals/lancet/article/PIIS0140-6736(20)32206-6/fulltext)), artificially inflating the fatality rate does not seem like the right solution. Ideally, aged care facilities would be modeled as a separate contact network. Failing that, the Prem matrices could be adjusted (or even calibrated) to account for higher contacts among older adults. As it is, this approach is almost certain to underestimate the number of infections among older adults, even if, by increasing the age-dependent mortality rate, the total number of deaths is roughly correct. I believe this issue is also compounded by the way the CDR is being estimated, which probably underestimates the number of cases early on, and overestimates it later. The underestimate of cases would also explain the need for a higher infection-fatality ratio (IFR). I would be more inclined to revisit the CDR estimate than assume that major international studies have gotten the age-dependent IFR wrong by a factor of 2.48.
- Line 107: "without needing to unrealistically manipulate the risk of transmission by cluster. Risk of infection in metropolitan areas was estimated to be up to double that of regional areas" -- I'm

confused, it sounds like the first sentence is saying that risk of transmission does not depend on cluster, while the second is saying it does?

- Line 112: "Interaction between populations of different clusters was low in the context of significant restrictions on movement between regions." -- This makes sense, but I do not recall seeing results on this in the manuscript.

- Line 115: "However, face coverings had a considerably greater effect on reversing the epidemic, which was observable due to the sharp transition in the extent of their use when they were mandated." -- If one looks at Fig. 2, it is true that the face covering mandate occurs immediately before the inflection point of the modeled curve. However, if one looks at the data instead, the inflection point appears to be more strongly associated with the stage 4 lockdown.

- Line 141: "The minimal information provided on seasonal forcing is likely attributable to our simulation period spanning less than four months and so covering a small proportion of the cycling period, such that the effect could represent other secular changes during the period modelled." -- I would have to agree with the authors on this point, and suggest that the seasonal forcing term be removed from the model.

- Line 147: "The finding was also not unexpected given the marked shift in population use of face coverings at this time and the timing of the policy change in late July relative to the dramatic reversal in case numbers occurring around one week later." -- If the authors are able to speculate, it would be interesting to know why a mask policy was not implemented far sooner, given the minimal economic and social disruption it causes compared to a lockdown (as they note), and since even in the US (which itself was several months behind East and South Asia), mask use had become official guidance by April (<https://www.cdc.gov/coronavirus/2019-ncov/prevent-getting-sick/cloth-face-cover-guidance.html>). To my knowledge, Australia was one of the last countries in the world to recommend mask use in public (though of course other countries have had challenges with compliance or mixed political messages). I would be very curious to see scenarios showing what would have happened if masks had been introduced earlier in the epidemic -- e.g., in late June or along with the stage 3 lockdown.

- Line 154: "The dramatic effect of each of these interventions on the epidemic trajectory is partly attributable to our implementation of these processes as applying to both the infectious cases and the exposed individual." -- I am not sure this is true; since this was a calibrated parameter, had it been implemented as a one-sided rather than two-sided intervention, the calibrated value would simply have been twice as large -- right?

- Line 162: "the posterior estimate of the infectiousness of asymptomatic cases suggested around threefold lower infectiousness per unit time." -- Apologies if I missed it, but I am unclear what data were able to constrain this parameter.

- Line 171: "Although this factor seems extreme, age-specific infection fatality rate estimates increase dramatically with age,[7] and it should be noted that the age-specific infection fatality rate parameters we used increase up to three-fold with each successive decade of age.[6] Therefore, an age distribution of the infected population that is one decade higher than that simulated would be expected to have a comparable effect" -- I appreciate the authors' acknowledgement of this point. However, a one-decade age shift is quite significant, and not supported by the data (especially at older ages, where the proportion of the population drops off quite dramatically, as the authors show in Fig. 1C). Furthermore, as the authors allude, the estimates by O'Driscoll already take into account age-dependent IFR, so further corrective factors should not be needed. However, the authors do raise

a valid point that they should perhaps extend their model age bins out to e.g. 90 years rather than 75.

- Line 243: from the YouGov repository linked to, I can only find Australia-wide responses to these questions, not Victoria-specific, with much lower levels of compliance (as would be expected). Are data for Victoria available from a different source?

- Line 246: If the YouGov data that the tan function was fitted to included responses from regional Victoria, then it probably wouldn't be appropriate to shift the curve in this way.

- Line 267: Were any constraints placed on mixing between adjacent vs. non-adjacent regions? One would expect much higher mixing between, say, South and Southeast Metro regions than between Gippsland and the Grampians. (A map of Victoria showing each region, perhaps including case counts, could be a nice addition to the supplement, but is by no means essential.)

- Figure 1: I am a bit surprised by the mobility data. As the authors are obviously well aware, Melbourne's lockdown meant that "most businesses have been shut, residents aren't allowed to leave their homes, except for essential reasons and for one hour of exercise, and an overnight curfew has been in effect" (<https://www.ctvnews.ca/world/how-draconian-are-melbourne-s-coronavirus-lockdown-measures-1.5105833>). To say that these measures result in only a ~40% drop in workplace mobility and a ~30% drop in non-workplace metropolitan mobility seems quite unlikely. And indeed, Apple mobility data (<https://covid19.apple.com/mobility>) suggests closer to a 70% drop over the same time period. I don't have an explanation to account for this large difference, but it gives me considerable doubt about the Google mobility data. Likewise, although they do not quantify the overall magnitude of the effect, eyeballing Fig. 5c from the Zachreson et al study on mobility in Melbourne (<https://royalsocietypublishing.org/doi/10.1098/rsif.2020.0657>), which uses Facebook data, also seems consistent with a >70% mobility reduction rather than a 30-40% one. I also find it surprising because many US cities, none of which implemented lockdowns as strict as Melbourne's, also saw mobility drops of up to 70% (<https://www.safegraph.com/data-examples/covid19-commerce-patterns>).

- Figure 4: This figure is somewhat difficult to read (i.e. multiple shadings on a non-white background). What is causing the wiggles in notifications for no face coverings?

- The following papers should probably be cited:

<https://www.medrxiv.org/content/10.1101/2020.11.16.20232843v1>,

<https://www.medrxiv.org/content/10.1101/2021.01.11.21249630v2>. I understand they may have been avoided as they are still preprints, but seem relevant nonetheless. Meanwhile, the only preprint cited (Scott et al) appears to be in press at MJA (<https://www.mja.com.au/journal/2020/modelling-impact-reducing-control-measures-covid-19-pandemic-low-transmission-setting>).

- The AuTuMN (and summer) libraries are clear and well documented, which is much appreciated. However, I was not able to easily find the code for the specific analyses included in this paper. It would be helpful to see this in order to be able to reproduce the results of this paper.

- It was not clear to me whether the modeling work described in this paper was used to inform policy at any point, or whether it was a purely retrospective study. If the former, it would be worthwhile describing what those influences were.

SUPPLEMENT

- Supplement, p. 8: "In the event that the infection fatality rate for an age group is greater than the total proportion hospitalised" -- To avoid this, it would be better to use IFR as a derived parameter rather than an input parameter, with the transition probabilities infected-symptomatic, symptomatic-hospitalised, hospitalised-critical, and critical-dead being the input parameters; IFR would then be the product of all of these rates.

- Supplement, p. 10 and Fig. S4: I am a little concerned about this implementation for CDR. First, it doesn't seem to take into account the number of people infected. There were various points last year when locations that had successfully achieved epidemic control (e.g., NSW, South Korea) were performing far fewer tests per capita than places with uncontrolled epidemics. What matters is not (so much) the number of tests per capita, but the number of tests per infection; test positivity rate, rather than tests per capita, is a better correlate of CDR. The true number of infections of course is unknown and must be modeled. However, using the number of tests per capita is simply not relevant. This can be seen in Fig. S4, where it seems highly improbable that the highest detection rates coincided with the peak of the epidemic, when contact tracers and the rest of the health system was most overstretched; nor is it probable that the tail end of the epidemic could have been controlled with only ~50% detection of symptomatic cases (which, considering 30-50% of infections are asymptomatic, means only 25-35% of all cases were diagnosed; and of course most transmission happens before a person becomes symptomatic and is diagnosed, etc). In addition, the use of per capita tests ignores the characteristics of contact tracing, which is effective precisely because it does not need to scale per capita but rather per case, and is highly effective at identifying even asymptomatic infections.

- Supplement, p. 13: It's not clear to me what is meant by "quadratically reducing the contribution of workplace contacts". Why quadratic? Why fit a spline rather than use the data directly? If the data are too noisy to use directly, why not use a smoothing kernel (e.g. Gaussian) rather than a spline?

- Supplement, p. 14: If data are available for increased time spent at home, why not use it? We know that infection risk is proportional to time spent in close proximity, so spending more time at home would be expected to increase the within-home transmission risk.

- Supplement, p. 17: For face coverings, I agree squaring the term makes sense, since one person could be wearing a mask in the interaction and the other might not be, i.e. $m(t) \propto f_1(t) \times f_2(t)$, where f_1 and f_2 are the probability that each person is wearing a mask. If you assume that these probabilities are equal (which of course one must in a compartmental model), and further assume masks have the same reduction on both transmission and acquisition, then I agree with $m(t) \propto f(t)^2$. However, this logic isn't the same for distancing: you can't have person 1 distancing but person 2 not distancing. Does this matter? Maybe not, although it does have behavioural implications: if I'm standing in a queue, wearing a mask vs. not reduces my risk of infection by f (not by f^2), but physically distancing with the person in front of me reduces my risk of infection by d^2 (not d).

- Supplement, p. 17: The choice of a hyperbolic (typo: "hyerbolic") tan function is unusual, but I can see why it was desirable (compared to a smoothing kernel or smoothing spline) in order to draw better inferences about the impact of each intervention. I feel like a bootstrapping or MCMC approach would be a better way to calculate the fits, but this is a very minor point.

- Supplement, p. 25: Given that almost all of the paper's results pertain to the calibrated parameter values, the calibration procedure is central to the paper's methodology, but is explained relatively briefly. Greater detail on the algorithm, such as how many iterations it was run for and with how many chains, plots of how goodness-of-fit (log-likelihood) improved over time, etc., would be valuable.

- Supplement, Figs. S7-S12: To me, the fits do not look especially convincing, though it's hard to tell

without e.g. 7-day moving average data to compare against. If one ignores the (very large) confidence intervals and instead focuses on the median, it seems quite a bit too low in many cases. Again, given the centrality of calibration to the paper, I find this concerning.

- Supplement, Figs. S13-S17: Given the quality of the fits to data, and given the priors, I'm surprised the posteriors are as "nicely behaved" as they are: almost all are Gaussians it seems. It does not seem there is enough data, or that the model dynamics are complex enough, to constrain so many parameters. Looking through the code (<https://github.com/monash-emu/AuTuMN/blob/master/autumn/calibration/calibration.py>) the most likely explanation I can see is that not enough chains are used and/or they are not run for long enough to ensure the chains are well-mixed, although I couldn't find the scripts where calibration is actually being run.

- Supplement, Figs. S18-S20: I am not quite sure what the point of these figures is; they seem to be showing noise (cf. infectious seed vs. other parameters; to me they look the same), and I couldn't find a reference to them in the text. At minimum, the x and y axes should be labeled.

- Supplement, Fig. S22: Can the legend be expanded, including the two colors of dot?

Reviewer #2 (Remarks to the Author):

It is good work with a thorough retrospective analysis of the effect of NPIs in the evolution of the Covid-19 pandemic in the Victoria region, and I'm positive about the publication of this paper.

A few remarks the author may want to discuss:

1. Unless I missed something, deaths are not considered in the counterfactual analysis. However, although we are looking at a short-medium period, the compromise between death toll/ closures' price is important to size future emergencies, the first being somehow proportional to the new cases, the second to the cumulative closure times in open/close strategies.

2. Use of facial masks is indeed important in containing the spread of infections. Differences are obvious whether masks are worn in open or close spaces. Is this taken into account by the seasonal parameter? Btw, the importance of this time-varying parameter is not fully clear in a short-time analysis (July-September if I understood well).

3. The role of asymptomatic individuals (and especially superspreaders) should be commented. Their role seems to be underestimated. They are not detected but there are probably estimates based on serological data. Actual and perceived CFR depend on the discrepancy between tested and non tested infected individuals.

4. The mathematical model is rather complex since stratified in space, age, activities. However, still is a networked compartmental model whose overall dynamics depends by classical parameters (as for instance the local R_t and networked R_t). All parameters are not fully identifiable, so I guess there are different combination of parameters that can give a certain R_t . Could you please discuss further this issue by quantifying the increase /decrease of R_t wrt the adopted containment measures?

5. Being the paper an analysis and interpretation of the past in a particular region, it would be important to stress what we can learn in order to drive political/social measures for possible future epidemics not affected by current vaccines and how far these containment measures are exportable to other regions.

In the current manuscript, authors developed and calibrated a compartmental mathematical model to epidemiological data to estimate the effectiveness of several non-pharmaceutical measures in containing the second covid-19 wave in Victoria, Australia. Overall their model fits well to the data and resultant estimates provide reasonable explanations about how combination of these interventions successfully curbed the second wave. I have some comments with respect to model and framework that are listed below:

1. Colors in the model diagram (Figure 1B) seem inconsistent with (Figure 1A). For example, why are the colors for the 3rd Ambulatory compartment not the same as the 2nd Infectious compartment in A. Only after reading SI, I can figure out that they represent the infectiousness of the compartment. It would be helpful to add that explanation to the figure caption in the main text too. In Figure 1B, saying Symptomatic instead of Symptomatic, ever detected should be sufficient.
2. Although there are figure captions. It would be useful if Figure is self explanatory with title and appropriate legends
3. Contact patterns were updated recently by Prem et a. 2017, and are available at <https://cmmid.github.io/topics/covid19/synthetic-contact-matrices.html>. While it's not peer reviewed yet, it would be useful to see if there is any significant changes for Australia and if incorporating them have any impact of your findings.
4. It is assumed in the model that Asymptomatic and Symptomatic ambulatory are never detected. Why is this assumed? Was no contact tracing and testing conducted in Australia? Or Was testing conducted only in individuals who showed more than mild symptoms?
5. Please add discussion about your reasoning including seasonal forcing. Usually seasonal forcing are good way to capture changing human contact patterns over the year. However, in the pandemic year with several social distancing measures dictated the contact patterns, I am not sure if it is a necessary component.
6. Why was the case detection rate based only on per capita tests conducted and not also positivity rate. As the positivity rate and necessary per capita test required to capture most cases, it would be more appropriate to incorporate it.
7. Who contributed to fatalities? Individuals who are in hospital and ICU? What was the average time to death for an individual in ICU and for someone in Hospital.

REVIEWER COMMENTS

Reviewer #1 (Remarks to the Author):

This study used a detailed age- and location-stratified compartmental model to explore the COVID-19 outbreak in Victoria, Australia, from June through October 2020. The key findings relate to the calibration of the model to the epidemic data via an MCMC algorithm, which allows the determination of parameter values, which are then interpreted in terms of their broader implications (e.g., the relatively high efficacy of face masks). The study draws on a wide variety of different data sources, including epidemiological data, mobility data, demographic data, etc. The findings are broadly consistent with previously published estimates, e.g. regarding face mask efficacy and other parameters.

MAJOR COMMENTS

This study has much to commend it, including the use of a carefully designed, open-source modeling library and the use of a large number of different data sources. However, I had a number of concerns regarding the methodology. Most seriously, almost all of the paper's results hinge on the calibration. A large number of parameters were simultaneously calibrated (20), despite the fact that relatively little data was available to calibrate to. While there were different regions, most regions had similar temporal trends, and experienced the same interventions at the same times (with some exceptions regarding regional vs. metro areas), so these regions are not really independent.

Many thanks for the positive comments. The repository structures have advanced further since the previous review (mostly github.com/monash-emu/autumn, rather than github.com/monash-emu/summer) and our application of a similar model to the Philippines has now been published with Lancet Regional Health Western Pacific, which is now mentioned in first section of the Supplement. The adaptation of our calibration algorithm is described below.

Put simply, the data comprise a noisy set of data points that comprise a near-symmetric curve. It is somewhat unclear to me how one could constrain reasonably well even 3-4 parameters with these data, much less 20 parameters (not to mention that these parameters were also combined with a time-varying 144x144 mixing matrix, which has its own considerable uncertainties associated with it). Not enough details are given about the calibration algorithm to independently validate it. Not only do the parameter distributions look curiously Gaussian, but there is much less degeneracy or cross-correlation between the parameters than one would expect. For example, couldn't one get identical results by multiplying each of the cluster-specific contact rates by a factor of 2 and then dividing the unadjusted risk of transmission by a factor of 2? Trade-offs such as these should both smear out the univariate parameter distributions and introduce (anti)correlations between them; the fact that these do not appear makes me think that parameter space was not explored sufficiently by the MCMC algorithm, rather than that the distributions truly are (almost) all well-constrained Gaussians.

In response to this major comment, we have revised our calibration approach to ensure it is as parsimonious as possible.

We agree broadly with the reviewer that a simpler calibration algorithm is likely to be more transparent and interpretable to readers and have favoured these principles in designing our revised algorithm. Although we see the advantages of this approach, we do not believe there was anything invalid about the previous algorithm. That is, it is not an essential aspect of calibration that all parameters that are varied through the algorithm are constrained. Rather, there are arguments for including additional calibration parameters that are unconstrained by data, such as:

- Propagating the genuine uncertainty associated with uncertain input parameters through the future projections
- Demonstrating that parameter values were unconstrained by data, which is potentially of interest (e.g. finding that the calibration process did not provide clear evidence for the strength of seasonal forcing is a conclusion that may be of some interest)

Nevertheless, we have fully redesigned the calibration process in a way that we believe is both more parsimonious and aligned with the reviewer's preferences, as described in **Table 1**.

Previous calibration parameter(s)	New calibration parameter(s)
Unadjusted risk of transmission per contact	Unchanged
Incubation period	Unchanged
Duration of active disease	Unchanged
Pre-ICU duration	Unchanged
Symptomatic proportion adjuster	Removed (note that the infectiousness of asymptomatic persons multiplier would have a similar effect and so captures the uncertainty around the importance of asymptomatic transmission)
Infection fatality rate adjuster	Applied only to the oldest age bracket, rather than to all age groups, as described below
Hospitalisation rate adjuster	Unchanged
Infectiousness of asymptomatic persons multiplier	Unchanged
Starting infectious population	Unchanged
Seasonal forcing	Removed from model
Case detection rate at one test per 1,000 per day (proportion)	Unchanged
Contact tracing efficiency at prevalence of one active case per 1,000 population	Newly included parameter, based on reviewer requests below
Inter-cluster mixing	Essentially unchanged, although now has a slightly different interpretation as described below
Effect of physical distancing	Unchanged
Effect of face coverings	Unchanged
Contact rate multiplier, North Metro	Reconciled into a single multiplier parameter for these two geographically contiguous regions
Contact rate multiplier, West Metro	
Contact rate multiplier, South Metro	
Contact rate multiplier, South East Metro	Reconciled into a single multiplier parameter for these two geographically contiguous regions
Contact rate multiplier, Barwon South West	Unchanged
Contact rate multiplier, regional services	Unchanged (single multiplier for all of the five regional services)

Table 1. Summary of changes to calibration approach compared to original submission. Changes associated with the removal of a parameter shaded in grey; change associated with the addition of a new parameter shaded in light blue.

To summarise, four parameters have been removed, one simplified and one new parameter added. This results in a calibration algorithm that has three fewer parameters for a more complicated model than in the original submission (with the elaboration of the model in relation to contact tracing described below).

To provide confidence that the parameter space has been fully explored, we ran the calibration for more than 10,000 iterations per chain (following optimisation of the run-time of our underlying model runner to allow for a greater number of iterations) and present further parameter diagnostics in the Supplement. While we recognise this is a potentially important consideration, we do not believe that there is any reason to suspect that our estimated parameter posteriors are incorrect because they are close to well-constrained Gaussian distributions. That is, we believe that the independence of some of the parameters is an interesting and valid finding from the calibration approach and suggests the parameters are constrained by different data sources.

We agree with the Reviewer that some parameter combinations were somewhat correlated and that different parameter sets may produce similar outputs. The correlations were already described in the original submission and we have included discussion of the fact the calibrated parameters are not fully identifiable.

With regards to the specific example of expected correlation between contact rate and the contact rate modifiers, Figure 1 of this document demonstrates that there is a moderate degree of correlation between higher contact rates and lower contact rate adjusters. This is more pronounced for the more highly affected services, which is exactly the result we would expect. We have added the following sentence to the Results section of the paper:

“Consistent with our intuition around these parameters, the location-specific adjusters to the contact rate were generally correlated with one another, but anti-correlated with the transmission risk parameter.”

Figure 1. Contact rate and contact rate modifier parameter correlation matrix.

Last, regarding the complexity of the mixing matrix, we also present a sensitivity analysis using an alternative mixing matrix derived from the Prem et al. pre-print from 2020.

There are several other methodological concerns, described in more detail below. In my view, most serious of these are (a) the way the case-detection rate was calculated (which does not seem to be correct in a context of low case numbers and active contact tracing); (b) the large correction factor required for the infection-fatality ratio, which to me suggests some methodological concerns; and (c) the use of Google mobility data, which does not seem to be consistent with other local mobility sources for Melbourne (e.g. Apple, Facebook), nor is it consistent with international sources (e.g. the claim that Melbourne, under strict lockdown, experienced a smaller drop in mobility than some US cities that had only weak mobility restrictions).

(a)

We acknowledge that the lack of model structure to capture contact tracing was an important limitation of the model and have now incorporated contact tracing into our model. This represents the largest change made to the model and the manuscript since the original submission and was a substantial piece of work, but significantly advances the computational structures we now have available to model.

The approach to simulating case detection remains unchanged and continues to assume that the proportion of symptomatic cases detected through passive presentation to health care services relates to the availability of testing, calculated as tests performed per population per day. However,

we have now included structure in the model to account for tracing of first order contacts of confirmed cases that saturates as the severity of the epidemic increases. Although this adds further to the complexity of the model, we believe this was probably the most important limitation of the previous model structure and was mentioned by two reviewers, such that the model is significantly strengthened by including this process.

We present the methods for this process in full in the Supplement, but briefly we stratified the model compartments representing active COVID-19 episodes (i.e. all compartments other than the susceptible and recovered compartments) according to contact tracing status (i.e. traced and untraced). Those patients who have been traced are considered less infectious than those that have not, representing quarantine of these persons. The process of contact tracing is simulated as a transition flow that moves a proportion of patients with COVID-19 (both symptomatic and asymptomatic) from the untraced to the traced stratum during their incubation period. The proportion of patients transitioning is calculated as the product of two proportions: 1) the symptomatic case detection rate, and 2) the proportion of contacts of known cases traced. The first proportion sets a ceiling for the overall proportion of infections being traced at the symptomatic case detection rate. The second proportion scales down with increasing prevalence to represent finite capacity of the health department to trace contacts as the epidemic worsens. Traced contacts with active COVID-19 are assumed to be detected and so contribute to notifications.

(b)

This is an important point that we have considered in more detail in light of this prompt and discuss as follows. The surveillance data to which we calibrated consist of 18,459 notifications and 801 deaths, for a case-fatality rate of 4.3%. (Of these notifications, 752 occurred in persons aged 65-74, 877 in persons aged 75-84 and 1,318 in persons aged 85 and above.) If we assume full case detection (i.e. $IFR=CFR$), then we would expect approximately 200 deaths, whereas 801 deaths were observed. Therefore, we need to consider explanations for this four-fold discrepancy, the simplest of which would be that around one quarter of all cases were detected, process 1). However, there are two further processes that could explain this differences: 2) a greater infection fatality rate in our setting, 3) concentration of the epidemic in older and more vulnerable groups within age strata. Each of these three processes is explicitly captured in our model in order to avoid having to inflate the infection fatality rate too far. The proportion of notifications in older age groups was already slightly greater than observed (around 20% rather than 16%), which was driven by our fixed input parameter that assumed the elderly are 1.41-fold more susceptible to infection than other adult age groups. Therefore, inflating this parameter further would seem unrealistic. Modelled case detection reached around 60 to 85% of symptomatic cases passively detected at maximum testing rates, while the proportion of contacts of identified index cases quarantined fell from 100% to around 30 to 60% at peak prevalence. We believe these figures are plausible and are varied across reasonable ranges during calibration. Allowing for much lower values would be inconsistent with our understanding of the epidemic (given the wide availability and compliance with testing for symptomatic episodes), and the (limited) serosurveillance data available. To address this comment, we have changed our assumption that the mortality risk should be inflated in the same way for all age groups and replaced this with the assumption that the mortality risk only increased in the oldest age group. This parameter now has a more intuitive interpretation (being an IFR for a specific age group, rather than an adjuster parameter) and reaches values that are both plausible and consistent with the discussion immediately above (8 to 30%). Given the concentration of the epidemic in residential aged care facilities, applying the adjustment to only the oldest age group is appropriate, given that most Victorian aged care residents are aged 75 and above.

(c)

This is addressed in detail below.

Finally, after crafting such a rich model, I was surprised that the authors placed so much emphasis on the calibration step, which is often -- if unfairly! -- regarded as a nuisance rather than "real" results. The scenarios section was intriguing, but was quite brief, and I felt many stones were left unturned in terms of (a) exploring how many infections and deaths could have been averted by an earlier face mask requirement, (b) comparing differences between regions (clusters), (c) exploring whether fits of similar log-likelihood could have been achieved with a much simpler model (e.g., apply the Bayesian information criterion), etc. The paper is already of reasonable length (especially with the supplement), and I am not suggesting that the authors make it even longer. However, I do wonder whether a more impactful paper could be written by taking empirical best-estimate parameters -- for both face mask efficacy and infection-fatality rates, for example -- at face value, instead of trying to calibrate them, and then use this as a more robust starting point to explore quantities that are truly unknown, such as the case detection rate at different points in the epidemic, or even the change in transmission due to spatiotemporal changes in mobility.

We agree that scenario projections from models are often the most interesting outcomes from studies like this. However, we would argue that it is even more important for readers to understand the reasons for the forward projections behaving in the way that they do. Therefore, we have included additional scenarios, while retaining a significant focus of the paper on understanding the simulation of the epidemic which underpins our projections. For example, the no face coverings counterfactual shows a very large epidemic, which is entirely dependent on the calibrated estimate of the parameter governing the effectiveness of this interventions.

Nevertheless, we have now run several additional scenarios in this revised version. These include the original scenarios that estimated the size of epidemics that would have occurred if mitigation measures had not been undertaken as well as the impact earlier responses could have had on reducing the size of the epidemic.

MINOR COMMENTS

- Line 21: *"The impacts of the virus [...]" -- I found this sentence a little confusing.*

Apologies. This was a poorly worded sentence. The previous text:

"The impacts of the virus are felt through the direct effect of the virus, particularly through its considerable risk of mortality following infection and likely substantial post-infection sequelae, but also through the extreme lockdown measures often needed to achieve control."

Has been revised to:

"The impacts of the epidemic are felt directly through infection-related mortality and post-infection sequelae, as well as through the often highly restrictive public health measures needed to achieve control."

- Line 33: *It would be helpful to explain to readers what each of these stages indicate; a reader unfamiliar with the Victorian context would not appreciate that "stages 3 and 4" translate to Melbourne experiencing "one of the world's longest and toughest lockdowns" (<https://www.bbc.com/news/world-australia-54654646>)*

The following sentence has been added to describe the severity of the lockdown:

“Stage 4 restrictions included business closures, restriction of restaurants and cafes to take-away services only, remote schooling, restriction of travel to a five kilometre radius, an 8pm curfew and reduction in public transportation, constituting one of the world’s longest and toughest lockdowns.”

A citation for a news report describing the measures employed has been added and the reviewer’s suggested news report describing the uniqueness of the measures has also been included.

- Line 40: *"the experience of Victoria’s second wave is virtually unique in that the pattern of substantial and escalating daily community cases was reversed following these policy changes, with elimination subsequently achieved in November" -- It's not clear to me what aspect of this is being flagged as unique; much of this is similar to the experiences of other countries that achieved (temporary) elimination, such as New Zealand, South Korea, Vietnam, etc.*

We have revised this sentence from the original:

“Indeed, the experience of Victoria’s second wave is virtually unique in that the pattern of substantial and escalating daily community cases was reversed following these policy changes, with elimination subsequently achieved in November.”

To the following:

“Indeed, Victoria’s second wave was virtually unique, in that these policy changes reversed substantial and escalating community cases rates and supported subsequent sustained elimination, which was achieved for several months from November 2020.”

It was challenging to encapsulate the factors that made Victoria’s experience unique, but we believe that this revised sentence achieves this. Victoria’s wave was far beyond the rates of community transmission yet seen in New Zealand, and considerably greater than Vietnam on a per capita basis. South Korea has experienced an epidemic with comparable rates to that of Victoria’s second wave, but has only achieved partial epidemic control and has experienced a number of exacerbations of transmission over recent months.

- Line 48: *It might help to explain what a "cluster" means here, since it is often used in contrast to widespread community transmission (e.g., the Cedar Meats cluster), rather than as a component of it.*

In the revised manuscript, we have changed from using “cluster” to abbreviate “health service cluster” to using “service”, and have indicated this approach at first usage.

- Line 62: *Is 2-5% per contact per day? Is this the same for households, workplaces, and the community?*

Yes, the unadjusted contact rate is the same for each location. Of course, this is scaled in several ways to account for the various processes captured by the model, but is equal for each location before scaling. We have added the words “in any location” to this sentence to ensure this point is clear.

- Line 65: *What data were used to constrain the geographical mixing? I did not see this described in the supplement.*

This was previously included in the Supplement, although we have now revised our approach to spatial mixing to implement an adjacency-based mixing matrix (which is static over time and is then combined with the dynamic age-specific mixing matrix to create the final mixing matrix, **Table 2** and included in the Supplement).

	Barwon South West	Gippsland	Hume	Loddon-Mallee	Grampians	North Metro	South East Metro	South Metro	West Metro
Barwon South West	R	0	0	0	M	M	0	0	M
Gippsland	0	R	M	0	0	0	M	M	0
Hume	0	M	R	M	0	M	M	0	0
Loddon-Mallee	0	0	M	R	M	M	0	0	M
Grampians	M	0	0	M	R	M	0	0	M
North Metro	M	0	M	M	M	R	M	0	M
South East Metro	0	M	M	0	0	M	R	M	0
South Metro	0	M	0	0	0	0	M	R	0
West Metro	M	0	0	M	M	M	0	0	R

Table 2. Adjacency-based between-service mixing matrix. Where 0 indicates no mixing between spatial patches, M is the calibrated inter-cluster mixing parameter, R populates the diagonal of the matrix and indicates the complement of the remaining values for each row/column (and so may take a different value in each cell in which it appears).

- Line 70: "However, the risk of hospitalisation (and hence ICU admission) and of death given infection were considerably greater than our age-specific prior estimates obtained from the literature. This likely reflects higher rates of exposure and infection in population groups at particularly high risk of adverse outcomes, including residents of aged care facilities." -- While the cause might be correct (possibly worth citing [https://www.thelancet.com/journals/lancet/article/PIIS0140-6736\(20\)32206-6/fulltext](https://www.thelancet.com/journals/lancet/article/PIIS0140-6736(20)32206-6/fulltext)), artificially inflating the fatality rate does not seem like the right solution. Ideally, aged care facilities would be modeled as a separate contact network. Failing that, the Prem matrices could be adjusted (or even calibrated) to account for higher contacts among older adults. As it is, this approach is almost certain to underestimate the number of infections among older adults, even if, by increasing the age-dependent mortality rate, the total number of deaths is roughly correct. I believe this issue is also compounded by the way the CDR is being estimated, which probably underestimates the number of cases early on, and overestimates it later. The underestimate of cases would also explain the need for a higher infection-fatality ratio (IFR). I would be more inclined to revisit the CDR estimate than assume that major international studies have gotten the age-dependent IFR wrong by a factor of 2.48.

As described above, we think it is unlikely that reducing the case detection rate would be sufficient to achieve this unless reduced to an implausibly low value. Rather than inflate the IFR for all age groups, we have now adjusted the IFR for the oldest age group only (i.e. ≥ 75 years). The absolute value of this proportion is now estimated through our calibration process.

“Victoria’s second wave is known to have had particularly dramatic effects on residents of aged care facilities and health care workers,²⁶ which we did not explicitly capture except by varying parameters relating to disease severity. The concentration in aged care was likely the main factor requiring us to inflate the international estimate for the infection fatality rate for those aged over 75 years. Our results suggest a markedly higher IFR in this group than that estimated from other settings, and is consistent with the high raw case-fatality rate of 4.3% in the data used for fitting (801 deaths, 18,459 notifications). This highlights the importance of risk factors and comorbid conditions on the estimated IFR, which likely underpin some of the dramatic increases in IFR with increasing age and are particularly concentrated in residents of aged care facilities. Our inflation of the age-specific estimates of the risk of hospitalisation given symptomatic COVID-19 are also consistent with a more severe epidemic, although hospital admission is driven by factors other than disease severity. These include infection control and workforce capacity and staff isolation requirements in residential aged care facilities, which were particularly important to this epidemic wave.”

Line 107: "without needing to unrealistically manipulate the risk of transmission by cluster. Risk of infection in metropolitan areas was estimated to be up to double that of regional areas" -- I'm confused, it sounds like the first sentence is saying that risk of transmission does not depend on cluster, while the second is saying it does?

Probably this confusion arose from our use of the word “only” earlier in the first sentence, which we intended to mean that the infectious seed was introduced only through the metropolitan services, rather than that this was the only change that was made to capture the effects. To clarify our intention, the sentence has now been re-worded to:

“The lower rates of COVID-19 observed in regional services were captured with the introduction of the infectious seed through the metropolitan services along with a modest and plausible change to the risk of transmission by geographical region.”

- Line 112: "Interaction between populations of different clusters was low in the context of significant restrictions on movement between regions." -- This makes sense, but I do not recall seeing results on this in the manuscript.

Here we are referring to our “inter-service mixing” parameter (previously “inter-cluster mixing”), which was around 1-2%. Because this parameter is specific to the local context, it is not possible to compare this to estimates from others. (The interpretation of this parameter has now changed, as described above, although the conclusion remains the same.)

- Line 115: "However, face coverings had a considerably greater effect on reversing the epidemic, which was observable due to the sharp transition in the extent of their use when they were mandated." -- If one looks at Fig. 2, it is true that the face covering mandate occurs immediately before the inflection point of the modeled curve. However, if one looks at the data instead, the inflection point appears to be more strongly associated with the stage 4 lockdown.

We agree and believe that this is entirely consistent with our findings. Specifically to look for a policy change that would explain the observed dramatic reversal in case rates, one should consider changes that occurred approximately one serial interval (or generation time) prior to the observed change. Further, it is also worth considering the delay to reporting of identified cases in this process. Therefore, we have now advanced all notification data one day earlier to allow for the average one day delay between testing and diagnosis/notification.

Due to the variation in case numbers, it is not possible to specify an exact date that the reversal occurred, but from the data it was likely in the last 2-3 days of July or the first 2-3 days of August. Given the modelled incubation and active periods, our modelled serial interval would be around five to seven days, such that with one day of delay from testing to notification, a policy change around the 23rd of July would be a highly plausible explanation for the reversal in cases. By contrast, the introduction of stage 4 restrictions would be expected to have impact around the second week of August, at which time case numbers had clearly begun to decline.

- Line 141: *"The minimal information provided on seasonal forcing is likely attributable to our simulation period spanning less than four months and so covering a small proportion of the cycling period, such that the effect could represent other secular changes during the period modelled." -- I would have to agree with the authors on this point, and suggest that the seasonal forcing term be removed from the model.*

We have removed seasonal forcing from the model. We agree with the reviewer that it is not necessary to include and also helps to reduce the parameter space as suggested in the first major comment. The reason for its conclusion was our general view that many settings appear to have observed some effect of seasonality (e.g. the general pattern of the European epidemics through 2020) and so we previously wished to allow for some effect of this process. Nevertheless it does not notably influence the other model conclusions and parameter estimates and so can be reasonably removed.

- Line 147: *"The finding was also not unexpected given the marked shift in population use of face coverings at this time and the timing of the policy change in late July relative to the dramatic reversal in case numbers occurring around one week later." -- If the authors are able to speculate, it would be interesting to know why a mask policy was not implemented far sooner, given the minimal economic and social disruption it causes compared to a lockdown (as they note), and since even in the US (which itself was several months behind East and South Asia), mask use had become official guidance by April (<https://www.cdc.gov/coronavirus/2019-ncov/prevent-getting-sick/cloth-face-cover-guidance.html>). To my knowledge, Australia was one of the last countries in the world to recommend mask use in public (though of course other countries have had challenges with compliance or mixed political messages). I would be very curious to see scenarios showing what would have happened if masks had been introduced earlier in the epidemic -- e.g., in late June or along with the stage 3 lockdown.*

Several additional scenarios have now been run. These include the effects of implementing earlier restrictions from the time of the implementation of stage 3 (9th July) and from the time implementation of the of postcode lockdowns (30th June).

- Line 154: *"The dramatic effect of each of these interventions on the epidemic trajectory is partly attributable to our implementation of these processes as applying to both the infectious cases and the exposed individual." -- I am not sure this is true; since this was a calibrated parameter, had it been implemented as a one-sided rather than two-sided intervention, the calibrated value would simply have been twice as large -- right?*

We agree with the reviewer that we are referring here to the size of the effect relative to the value of the parameter and have added the following text: "relative to the parameter estimate that

suggests partial efficacy". Although the parameter value could have been scaled if it was applied directly, it is not a matter of multiplying the parameter by two for a number of reasons. These include that the effect is calculated as $(1 - \text{old_param})^2$, whereas if a simple scaling parameter were used it would be applied as $1 - \text{new_param}$. However, it would still not be a matter of solving for new_param , because the effect is only applied to certain locations, so the actual model process is more complex. Nevertheless, the principle is correct that applying the parameter as a direct modifier rather than with our quadratic approach would have led to a larger estimated effect.

- Line 162: *"the posterior estimate of the infectiousness of asymptomatic cases suggested around threefold lower infectiousness per unit time." -- Apologies if I missed it, but I am unclear what data were able to constrain this parameter.*

This parameter was included because it is highly uncertain and we were particularly keen to allow for uncertainty in the characteristics of undetected cases. The value of the parameter appeared moderately constrained by the data, although it was fairly broadly distributed across the range of its prior distribution. Therefore, we have not based any major conclusions on the inferences around this parameter.

The processes that could constrain this value could include that changes in detection of symptomatic cases will have a stronger effect if asymptomatic cases are assumed never to be detected. Therefore, the increase in testing and consequently in CDR during July may have a greater effect on reversing the epidemic trajectory if this parameter is low. We considered these arguments too speculative to include in the manuscript.

- Line 171: *"Although this factor seems extreme, age-specific infection fatality rate estimates increase dramatically with age,[7] and it should be noted that the age-specific infection fatality rate parameters we used increase up to three-fold with each successive decade of age.[6] Therefore, an age distribution of the infected population that is one decade higher than that simulated would be expected to have a comparable effect" -- I appreciate the authors' acknowledgement of this point. However, a one-decade age shift is quite significant, and not supported by the data (especially at older ages, where the proportion of the population drops off quite dramatically, as the authors show in Fig. 1C). Furthermore, as the authors allude, the estimates by O'Driscoll already take into account age-dependent IFR, so further corrective factors should not be needed. However, the authors do raise a valid point that they should perhaps extend their model age bins out to e.g. 90 years rather than 75.*

These general issues are addressed in the earlier response regarding IFR estimates. Extending to additional age groups would be very difficult to do because of lack of data on the age-specific population distribution, parameter values and (especially) mixing matrices.

- Line 243: *from the YouGov repository linked to, I can only find Australia-wide responses to these questions, not Victoria-specific, with much lower levels of compliance (as would be expected). Are data for Victoria available from a different source?*

Victoria-specific data are available from this repository at <https://github.com/YouGov-Data/covid-19-tracker/blob/master/data/australia.zip> (downloaded 24th June 2021). The Victoria-specific estimates were used.

- *Line 246: If the YouGov data that the tan function was fitted to included responses from regional Victoria, then it probably wouldn't be appropriate to shift the curve in this way.*

Unfortunately YouGov survey data are not available at any finer resolution than at state level. As 80.1% of Victoria's population reside in the metropolitan services and we suspect that these surveys over-sample Metropolitan Melbourne, we believe that the predominant signal is coming from the policy change for Melbourne. This would be slightly diluted by some survey responses from regional Victoria, such that the true change in face covering use may have been even more dramatic than that seen through the data. However, there is no way to account for this exactly and the change over time is already extremely dramatic – from very limited usage to almost complete usage. Therefore, adjusting the function to make magnitude of the change slightly more extreme is very unlikely to change any of our overall conclusions.

For regional Victoria, we believe that the assumption that face covering use changed in an equivalent way to that for Metropolitan Melbourne is the most realistic assumption available to us. The only alternative assumptions that we could use from the data available would be that regional Victoria followed the pattern observed in Metropolitan Melbourne despite face coverings not being mandated until ten days later. This would be inconsistent with our local understanding of the epidemic and the transition from limited usage to full usage has been independently confirmed from Australian Government Department of Health survey data for regional Victoria specifically. Unfortunately the Department of Health data begin from late July and so cannot be used for curve fitting for Metropolitan Melbourne because they start from around the time of the policy change in the central area. We have not sought permission to share a plot of these data, which would require permissions that would considerably delay the progress of this manuscript. However, they show a very similar pattern for regional Victoria to that observed from the YouGov data ten days earlier, except with few data points for the pre-face covering period in late July. The paucity of the pre-policy change data even for Regional Victoria in this alternative dataset would also preclude accurate fitting of a function even if we were able to obtain permission to use this.

Therefore, having reviewed all the datasets available to us, we believe that our approach is the most appropriate available. We acknowledge the limitations, which we believe would make a minor to negligible difference to our conclusions.

We have added the following sentence to the limitation paragraph of the Discussion:

“Our assumption that the profile of face covering use in regional Victoria followed the policy change in an equivalent way to that of Metropolitan Melbourne cannot be verified from other publicly available sources.”

- *Line 267: Were any constraints placed on mixing between adjacent vs. non-adjacent regions? One would expect much higher mixing between, say, South and Southeast Metro regions than between Gippsland and the Grampians. (A map of Victoria showing each region, perhaps including case counts, could be a nice addition to the supplement, but is by no means essential.)*

We now model geographical interactions through an adjacency-based spatial mixing matrix, as described above (**Table 2**). We agree that direct movement from one spatial patch to a second non-contiguous patch would have occurred at negligible rates during the period simulated, which is a feature of this matrix.

- *Figure 1: I am a bit surprised by the mobility data. As the authors are obviously well aware, Melbourne's lockdown meant that "most businesses have been shut, residents aren't allowed to leave their homes, except for essential reasons and for one hour of exercise, and an overnight curfew has been in effect" (<https://www.ctvnews.ca/world/how-draconian-are-melbourne-s-coronavirus-lockdown-measures-1.5105833>). To say that these measures result in only a ~40% drop in workplace mobility and a ~30% drop in non-workplace metropolitan mobility seems quite unlikely. And indeed, Apple mobility data (<https://covid19.apple.com/mobility>) suggests closer to a 70% drop over the same time period. I don't have an explanation to account for this large difference, but it gives me considerable doubt about the Google mobility data. Likewise, although they do not quantify the overall magnitude of the effect, eyeballing Fig. 5c from the Zachreson et al study on mobility in Melbourne (<https://royalsocietypublishing.org/doi/10.1098/rsif.2020.0657>), which uses Facebook data, also seems consistent with a >70% mobility reduction rather than a 30-40% one. I also find it surprising because many US cities, none of which implemented lockdowns as strict as Melbourne's, also saw mobility drops of up to 70% (<https://www.safegraph.com/data-examples/covid19-commerce-patterns>).*

We are not surprised by the mobility estimates, which are entirely consistent other data sources.

Although Victoria's lockdown was considered quite stringent by comparison to many other settings internationally, various activities continued that have been restricted in other settings. For example, cafes continued to operate for take-away services, exercise was permitted and public transport continued with reduced timetabling.

More importantly for the absolute estimates, the Google mobility data are entirely consistent with data streams from other sources.

First regarding Facebook data, the following graph presents data for "movement range" defined as the number of map tiles a person visited in a day, relative to baseline estimates from February 2020 (available at <https://insightplus.mja.com.au/2020/41/what-mobility-data-can-tell-us-about-covid-19-lockdowns/>). Blue represents Melbourne local government areas and orange other Victorian local government areas. As these data do not distinguish by location (i.e. home/workplace/school/other), a decrease to 40 to 50% of baseline values in Metropolitan Melbourne when stage 4 was implemented would be entirely consistent with our approach which applies a decrease to 50 to 60% of baseline in workplaces and other locations with the complete closure of schools.

Similarly for Apple data, the nadir mobility values for Melbourne reached around 40% of baseline values for most travel modalities, which would again be consistent with our estimates.

It is not possible to revise our approach to incorporating Google mobility data, which is fundamentally dependent on the ability to map from the Google mobility locations (and policies on school closures) to the contact locations used in the Prem matrices. This is the major advantage of the Google mobility data, which is not available from these other data sources.

- Figure 4: This figure is somewhat difficult to read (i.e. multiple shadings on a non-white background). What is causing the wiggles in notifications for no face coverings?

The reason for the wiggles in notifications is that the shaded areas are quantiles over multiple model runs at a particularly point in time. Therefore, there is some randomness to the point at which an individual run will peak, resulting in variation in the median estimate as various curves peak at times close to one-another to create small variations in the shaded credible intervals. The previous changes were not attributable to parameter changes. In the revised version, we have run the model for a greater number of iterations, but also have a more complex model in which parameter values also change in response to the epidemic trajectories.

The shading colours of Figure 4 have been revised to improve clarity.

- The following papers should probably be cited:

<https://www.medrxiv.org/content/10.1101/2020.11.16.20232843v1>,

<https://www.medrxiv.org/content/10.1101/2021.01.11.21249630v2>. I understand they may have been avoided as they are still preprints, but seem relevant nonetheless.

Meanwhile, the only preprint cited (Scott et al) appears to be in press at MJA

(<https://www.mja.com.au/journal/2020/modelling-impact-reducing-control-measures-covid-19-pandemic-low-transmission-setting>).

We agree these are important and relevant works. References have been added to discuss these studies in relation to our findings. We refer to the first citation to support our finding that an earlier lockdown would likely have significantly reduced the epidemic final size. We refer to the second citation (Blakely et al.) to emphasise that part of the purpose of an aggressive lockdown towards elimination may actually be to reduce the overall need for lockdowns over a longer period.

- The AuTuMN (and summer) libraries are clear and well documented, which is much appreciated. However, I was not able to easily find the code for the specific analyses included in this paper. It would be helpful to see this in order to be able to reproduce the results of this paper.

The following git tag links to the exact code from the AuTuMN repository that was used for the simulations run in this revision: https://github.com/monash-emu/AuTuMN/releases/tag/vic_revision_nature_comms

- It was not clear to me whether the modeling work described in this paper was used to inform policy at any point, or whether it was a purely retrospective study. If the former, it would be worthwhile describing what those influences were.

The authors affiliated to the university where the modelling was primarily undertaken (name withheld for blinded peer review) provided projections of specific policy interventions, including school closures and elderly protection. One model for each health service cluster was run weekly (i.e. without the geographical structure we use in this analysis) to provide projections of future epidemic burden from the second-wave period modelled through to the end of December 2020. These analyses were undertaken in close collaboration with several Departmental staff, including co-authors to this paper, and became the official Victorian Government projections. As intellectual

property of the Victorian Government, they were then available for use across government departments as required. The first sentence of the Methods section has been extended to the following to describe the exact use of the model by the Victorian Government.

“We adapted the transmission dynamic model that was previously used to produce policy-relevant analyses and projections of specific public health and social measures in 2020. The model was also used to forecast new cases, health system capacity requirements and deaths for the Victorian Department of Health and Human Services (DHHS) and the Government of Victoria at the health service cluster level (henceforward “service”) during the second wave to December 2020.”

SUPPLEMENT

- Supplement, p. 8: "In the event that the infection fatality rate for an age group is greater than the total proportion hospitalised" -- To avoid this, it would be better to use IFR as a derived parameter rather than an input parameter, with the transition probabilities infected-symptomatic, symptomatic-hospitalised, hospitalised-critical, and critical-dead being the input parameters; IFR would then be the product of all of these rates.

We do not see a particular advantage to changing to this suggested approach and we believe it is reasonable to assume that deaths occurring in excess of hospitalisation rates would occur in the community. The implementation approach we have chosen has the further advantage of allowing any combination of age-specific IFRs to be requested, which we believe is more intuitive and manageable from a modelling perspective than attempting to allow these to emerge from the simulations.

Further, this approach is fundamental to the simulations we have run in Victoria and the multiple other settings to which this model has been applied for policy (e.g. the Philippines, Malaysia and Sri Lanka), such that it is unfortunately not practical to change this.

- Supplement, p. 10 and Fig. S4: I am a little concerned about this implementation for CDR. First, it doesn't seem to take into account the number of people infected. There were various points last year when locations that had successfully achieved epidemic control (e.g., NSW, South Korea) were performing far fewer tests per capita than places with uncontrolled epidemics. What matters is not (so much) the number of tests per capita, but the number of tests per infection; test positivity rate, rather than tests per capita, is a better correlate of CDR. The true number of infections of course is unknown and must be modeled. However, using the number of tests per capita is simply not relevant. This can be seen in Fig. S4, where it seems highly improbable that the highest detection rates coincided with the peak of the epidemic, when contact tracers and the rest of the health system was most overstretched; nor is it probable that the tail end of the epidemic could have been controlled with only ~50% detection of symptomatic cases (which, considering 30-50% of infections are asymptomatic, means only 25-35% of all cases were diagnosed; and of course most transmission happens before a person becomes symptomatic and is diagnosed, etc). In addition, the use of per capita tests ignores the characteristics of contact tracing, which is effective precisely because it does not need to scale per capita but rather per case, and is highly effective at identifying even asymptomatic infections.

We acknowledge this issue is important and have addressed this by including new structure for case diagnosis through contact tracing, as described above. We note that some of the quantities

suggested are epidemiological observations that cannot easily be projected forwards. For example, in projecting a future epidemic, it would be impossible to predict what the test positivity rate would be as case numbers climbed. We believe our revised approach has the advantages that it is both realistic and can be used for forward projections. This is described in more detail in response to Reviewer #3 below.

- Supplement, p. 13: It's not clear to me what is meant by "quadratically reducing the contribution of workplace contacts". Why quadratic? Why fit a spline rather than use the data directly? If the data are too noisy to use directly, why not use a smoothing kernel (e.g. Gaussian) rather than a spline?

By quadratic, we mean that the contacts are treated as pairs hence any mobility effect reducing population movement to a particular location is squared in the final model, by reducing both the density of the infected and the density of the susceptible individuals. The Supplement provides the equations to indicate this mathematically and the revised text of the manuscript should now make this clearer.

- Supplement, p. 14: If data are available for increased time spent at home, why not use it? We know that infection risk is proportional to time spent in close proximity, so spending more time at home would be expected to increase the within-home transmission risk.

To respond to this comment, we applied residential Google mobility to scale the home-based contacts of the Prem matrices in the revised model as described above. Keeping the model configuration otherwise unchanged, this resulted in posterior distributions of the incubation period of 2 to 3.5 days and of the duration of active disease around 4 to 5.5 days (i.e. in the extreme lower tail of the prior distributions for this parameter). Exploring the model through manual calibration, we determined that the explanation for this behaviour was that the effective reproduction number could only be reduced to just below one through the other time-varying processes if residential contacts were considered to increase towards the peak of the epidemic as people increasingly stayed at home. Because of this, the model could only recover the August epidemic downslope by markedly shortening the serial interval.

We present this alternative model configuration as a sensitivity analysis to demonstrate that this approach was less realistic, although it still returned similar posteriors with regards to our main parameters of interest.

It should also be noted that residential Google mobility is an estimate of time spent in residential locations rather than number of visits. This is because nearly everyone visits their own household every day, such that number of residential visits provides little information.

- Supplement, p. 17: For face coverings, I agree squaring the term makes sense, since one person could be wearing a mask in the interaction and the other might not be, i.e. $m(t) \propto f_1(t) \times f_2(t)$, where f_1 and f_2 are the probability that each person is wearing a mask. If you assume that these probabilities are equal (which of course one must in a compartmental model), and further assume masks have the same reduction on both transmission and acquisition, then I agree with $m(t) \propto f(t)^2$. However, this logic isn't the same for distancing: you can't have person 1 distancing but person 2 not distancing. Does this matter? Maybe not, although it does have behavioural implications: if I'm standing in a queue, wearing a mask vs. not reduces my risk of

infection by f (not by f^2), but physically distancing with the person in front of me reduces my risk of infection by d^2 (not d).

We chose our approach for consistency with the other implementations and also believe it is this most intuitive way to address this. Although the survey data on which the physical distancing function are based relate to that specific behaviour, they are likely to be a surrogate for a range of inter-personal behaviours that may reduce transmission. These behaviours are likely to have a greater effect if adopted by both persons potentially coming into contact. Further, if one person who consistently avoids two metres proximity comes into contact with another such person, we believe that the rate of contact would be lower than if one such person came into contact with another person who does not consistently avoid such contact. We also believe it would be considerably harder to explain the intuition for these effects if they were implemented in different ways within a single model.

- Supplement, p. 17: The choice of a hyperbolic (typo: "hyerbolic") tan function is unusual, but I can see why it was desirable (compared to a smoothing kernel or smoothing spline) in order to draw better inferences about the impact of each intervention. I feel like a bootstrapping or MCMC approach would be a better way to calculate the fits, but this is a very minor point.

The choice of a hyperbolic tan function provides a very good fit to the data with a minimum of parameters to estimate, each associated with a clear intuition. That is, the function requires parameters to define the upper limit, the lower limit and the inflection point, with one further parameter to define the shape (essentially the sharpness of the change from the lower to the upper estimate). A logistic function, which is a special case of the hyperbolic tan function group, would have been an acceptable alternative, but would not have any particular advantage over the functional form we chose.

Note that the hyperbolic tan function describes the shape of the function rather than the fitting process.

The typo has been corrected.

- Supplement, p. 25: Given that almost all of the paper's results pertain to the calibrated parameter values, the calibration procedure is central to the paper's methodology, but is explained relatively briefly. Greater detail on the algorithm, such as how many iterations it was run for and with how many chains, plots of how goodness-of-fit (log-likelihood) improved over time, etc., would be valuable.

Further diagnostics of the calibration algorithm are now presented in the Supplement. In particular, we have run all seven chains for more than 10,000 iterations after discarding burn-ins and provide a table of convergence statistics, demonstrating excellent convergence of the chain. We present several parameter correlation matrices, which demonstrate relationships between input parameters that are as would be anticipated given our intuition about their effects on the epidemic.

- Supplement, Figs. S7-S12: To me, the fits do not look especially convincing, though it's hard to tell without e.g. 7-day moving average data to compare against. If one ignores the (very large) confidence intervals and instead focuses on the median, it seems quite a bit too low in many cases. Again, given the centrality of calibration to the paper, I find this concerning.

The revised calibrations allow for a greater number of iterations and we believe now address these concerns.

- *Supplement, Figs. S13-S17: Given the quality of the fits to data, and given the priors, I'm surprised the posteriors are as "nicely behaved" as they are: almost all are Gaussians it seems. It does not seem there is enough data, or that the model dynamics are complex enough, to constrain so many parameters. Looking through the code (<https://github.com/monash-emu/AuTuMN/blob/master/autumn/calibration/calibration.py>) the most likely explanation I can see is that not enough chains are used and/or they are not run for long enough to ensure the chains are well-mixed, although I couldn't find the scripts where calibration is actually being run.*

Please see first response to main concerns above for a full response to the issues around calibration.

- *Supplement, Figs. S18-S20: I am not quite sure what the point of these figures is; they seem to be showing noise (cf. infectious seed vs. other parameters; to me they look the same), and I couldn't find a reference to them in the text. At minimum, the x and y axes should be labeled.*

They show the parameter values over sequential iterations of the model and confirm good mixing of chains. The y-axes were previously labelled. The x-axes have now been labelled. These figures have been retained because in responding to the reviewer's other requests we have generally favoured increasing the number of figures and visualisations to provide insights into the calibration process.

- *Supplement, Fig. S22: Can the legend be expanded, including the two colors of dot?*

The scenarios have now been entirely revised and are presented in the main manuscript. There was only one dot colour in this figure, but the shading appeared different because some dots were behind the shaded area in the z-order of the plot. This has now been fixed and the target data have been moved to the top layer of the plot.

Reviewer #2 (Remarks to the Author):

It is good work with a thorough retrospective analysis of the effect of NPIs in the evolution of the Covid-19 pandemic in the Victoria region, and I'm positive about the publication of this paper.

A few remarks the author may want to discuss:

1. Unless I missed something, deaths are not considered in the counterfactual analysis. However, although we are looking at a short-medium period, the compromise between death toll/ closures' prise is important to size future emergencies, the first being somehow proportional to the new cases, the second to the cumulative closure times in open/close strategies.

Deaths were previously presented as a panel of Figure 10.

2. Use of facial masks is indeed important in containing the spread of infections. Differences are obvious whether masks are weared in open or close spaces. Is this taken into account by the seasonal parameter? Btw, the importance of this time-varying parameter is not fully clear in a short-time analysis (July-September if I understood well).

Following suggestions from all reviewers, we have removed the seasonal forcing parameter. We agree that this is unlikely to have a significant effect on our model projections, as was demonstrated in the previous submission.

Note that we apply the effect of face coverings to non-household locations only, consistent with the policy. The Prem matrices are derived from diary reports of interactions in certain settings, which are usually indoor, and so emphasises the importance of indoor contacts to transmission. This is fundamental to our approach and is likely to be realistic, given the importance of indoor transmission to the COVID-19 epidemic.

3. The role of asymptomatic individuals (and especially superspreaders) should be commented. Their role seems to be underestimated. They are not detected but there are probably estimates based on serological data. Actual and perceived CFR depend on the discrepancy between tested and non tested infected individuals.

The revised model now considers asymptomatic persons with greater realism through the contact tracing structure. These individuals can now be detected and quarantined before infectiousness develops. The decrease in contact tracing efficiency ensures that asymptomatic persons become more important to model dynamics as the epidemic worsens. Variation in infectiousness is captured to some extent through the different levels of infectiousness assigned to different pathways through the clinical stratification of the model.

We reviewed results from <https://serotracker.com/en/Explore> to identify relevant serosurveys, with two that were relevant to Victoria identified. Serological data remain limited, but available data are consistent with our results. We have added the following paragraph to the Discussion section:

“Australia has experienced a minimal COVID-19 epidemic throughout 2020 and relatively few serosurveys have been undertaken in the country. A national serosurvey undertaken in elective surgery patients in four states including Victoria in June/July 2020 identified no seropositive patients from 3,037 samples tested, supporting our approach of commencing our simulations with a fully susceptible population.² A serosurvey of health care workers from Eastern Health in Melbourne’s eastern suburbs found a seropositive proportion of 2.2% in November/December 2020 in a group at higher risk of exposure than the general community.³ This is consistent with our estimates of a population-wide recovered proportion of around 1%, with marked differences by age group and location.”

4. The mathematical model is rather complex since stratified in space, age, activities. However, still is a networked compartmental model whose overall dynamics depends by classical parameters (as for instance the local R_t and networked R_t). All parameters are not fully identifiable, so I guess there are different combination of parameters that can give a certain R_t . Could you please discuss further this issue by quantifying the increase /decrease of R_t wrt the adopted containment measures?

We hope that the revised calibration approach is clearer and facilitates understanding of the parameter posteriors estimated. The parameter correlation matrices we presented address the issue of identifiability.

The macro-distancing and micro-distancing processes scale the effect of interventions with the square of the complement of the parameter value. Text has been added to the Discussion to clarify this: “i.e. the scaled transmission rate is the square of the complement of the intervention effect”. The scaling of the mobility data are presented in Figure 1 of the manuscript and the scaling of the

physical distancing and face coverings compliance functions are presented in the Supplement, which illustrates the effect of each process.

5. Being the paper an analysis and interpretation of the past in a particular region, it would be important to stress what we can learn in order to drive political/social measures for possible future epidemics not affected by current vaccines and how far these containment measures are exportable to other regions.

The last two sentences of the Conclusions paragraph of the Discussion section have been revised to emphasise the importance of public health and social measures to epidemic control.

Reviewer #3 (Remarks to the Author):

In the current manuscript, authors developed and calibrated a compartmental mathematical model to epidemiological data to estimate the effectiveness of several non-pharmaceutical measures in containing the second covid-19 wave in Victoria, Australia. Overall their model fits well to the data and resultant estimates provide reasonable explanations about how combination of these interventions successfully curbed the second wave. I have some comments with respect to model and framework that are listed below:

1. Colors in the model diagram (Figure 1B) seem inconsistent with (Figure 1A). For example, why are the colors for the 3rd Ambulatory compartment not the same as the 2nd Infectious compartment in A. Only after reading SI, I can figure out that they represent the infectiousness of the compartment. It would be helpful to add that explanation to the figure caption in the main text too. In Figure 1B, saying Symptomatic instead of Symptomatic, ever detected should be sufficient.

Thank you for these comments, which we agree with. We have adapted the caption to indicate that Figure 1A provides only example infectiousness shading and explained the shading of Figure 1B in the caption. Labelling the first callipers as “Symptomatic, ever detected” was actually previously incorrect, and these brackets should indeed have been marked “Symptomatic”. This has been corrected.

2. Although there are figure captions. It would be useful if Figure is self explanatory with title and appropriate legends

The caption to Figure 1 has been fully revised.

3. Contact patterns were updated recently by Prem et al. 2017, and are available at <https://cmmid.github.io/topics/covid19/synthetic-contact-matrices.html>. While it's not peer reviewed yet, it would be useful to see if there is any significant changes for Australia and if incorporating them have any impact of your findings.

This has been run as a sensitivity analysis, with similar posterior estimates for our main parameters of interest.

4. It is assumed in the model that Asymptomatic and Symptomatic ambulatory are never detected. Why is this assumed? Was no contact tracing and testing conducted in Australia? Or Was testing conducted only in individuals who showed more than mild symptoms?

As introduced above and explained in detail in the revised Supplement, we agree that this was previously the biggest limitation of the model. Structure to capture contact tracing that declines in efficiency with epidemic severity has now been introduced.

5. Please add discussion about your reasoning including seasonal forcing. Usually seasonal forcing are good way to capture changing human contact patterns over the year. However, in the pandemic year with several social distancing measures dictated the contact patterns, I am not sure if it is a necessary component.

As mentioned above, we have now removed the seasonal forcing structure from the model.

6. Why was the case detection rate based only on per capita tests conducted and not also positivity rate. As the positivity rate and necessary per capita test required to capture most cases, it would be more appropriate to incorporate it.

We considered various ways to capture case detection rate in the model before arriving at our current approach. Although we have not changed the approach, we believe that our new structure for contact tracing now much more accurately reflects the true epidemic dynamics.

We previously considered using positivity rate. However, there are many competing issues contributing to the proportion of all tests positive. At the start of an epidemic with very low case numbers, test positivity will often be low, but then increase with prevalence. In response, testing numbers will often be increased, leading to a decline in the positivity rate. Therefore, there is no consistent relationship between positivity rate and case detection rate.

Another important consideration is while we can consider that testing numbers remain constant in future projections, the assumption that test positivity remains constant in future projections regardless of epidemic size would be entirely invalid. Therefore, future projections would be impossible without deriving a relationship between epidemic size and test positivity from which CDR could in turn be derived. We believe that attempting to accurately capture this relationship would lead to a considerably less elegant model and make it difficult for readers to understand modelled dynamics.

Please note that under the revised modelling approach, persons diagnosed through contact tracing do contribute to notifications, such that the modelled case detection rate should now be conceptualised as the case detection rate for symptomatic persons passively presenting.

7. Who contributed to fatalities? Individuals who are in hospital and ICU? What was the average time to death for an individual in ICU and for someone in Hospital.

This is described in the Supplement and is one of the more complicated parts of our code. However it should be noted that it is not important which clinical strata contribute to the fatalities, except in determining the timing of the deaths. This is because we adapt the specific model parameters and flow rates to ensure we capture the desired IFR for a particular clinical stratum.

Most or all deaths occur in the hospitalised or ICU-admitted strata. The durations in hospital and ICU are 11.6 and 7.4 days, consistent with local observations last year. The time prior to ICU admission is calibrated, to allow better fitting to the profile of ICU admissions and deaths.

References

1. Cousins, S. Experts criticise Australia's aged care failings over COVID-19. *The Lancet* **396**, 1322-1323 (2020).
2. Coatsworth, N., *et al.* Prevalence of asymptomatic SARS-CoV-2 infection in elective surgical patients in Australia: a prospective surveillance study. *ANZ Journal of Surgery* **91**, 27-32 (2021).
3. Lau, J.S.Y., *et al.* SARS-CoV-2 seroprevalence in healthcare workers in a tertiary healthcare network in Victoria, Australia. *Infection, Disease & Health* (2021).

REVIEWER COMMENTS

Reviewer #1 (Remarks to the Author):

Overall, the paper is enormously improved and the authors are thanked for producing an exceptionally thorough response and revision. I have only a few remaining comments:

1. "That is, it is not an essential aspect of calibration that all parameters that are varied through the algorithm are constrained." -- This is true, but with the caveat that many of these parameters are constrained from empirical studies in other contexts. The "correct" approach is of course to use these empirical estimates as the prior.
2. "That is, we believe that the independence of some of the parameters is an interesting and valid finding from the calibration approach and suggests the parameters are constrained by different data sources." -- The way to test this would be to generate surrogate data using AuTuMN using several different parameter sets, add some noise, and then use the algorithm to try to recover the original parameter values -- i.e., test the identifiability of the model. If not too onerous, I would suggest doing this analysis as I still have some concerns about this aspect of the methodology. Note that even fairly simple models can have unidentifiable parameters (<https://www.ncbi.nlm.nih.gov/pmc/articles/PMC7752088/>). For parameters that are determined to be unidentifiable, I appreciate the authors' comment about propagating uncertainty, and here a prior distribution, rather than the posterior, may be as good (or even better) to use.
3. "However, we would argue that it is even more important for readers to understand the reasons for the forward projections behaving in the way that they do." -- I agree; the comment was more about whether it is really possible to get a more accurate estimate of, e.g., face mask efficacy from calibrating this particular model, rather than using best estimates from the empirical literature.
4. "It was challenging to encapsulate the factors that made Victoria's experience unique" -- Up to the authors to choose what's best (including leaving as-is), but personally I feel it would be more helpful to readers to also acknowledge the similarities between Victoria and other highly successful locations (e.g., Singapore; potentially Taiwan, although they have not yet achieved complete elimination of their most recent outbreak).
5. The supplement still uses the terminology "cluster", which I believe has been adapted to "service" in the main text. (Personally I still find these terms confusing if they refer predominantly just to "regions", but I defer to the authors on this point.) I also don't see expanded description of the calibration algorithm (e.g., the 10,000 iterations -- and was a single chain used for each cluster/service?).

Reviewer #2 (Remarks to the Author):

no more comments

Reviewer #3 (Remarks to the Author):

I thank the authors for addressing my comments as well as other reviewers. I congratulate the authors for doing a well-conducted study. Thanks

We thank the reviewers for their careful review of our revised study. Given the rapidly escalating situation in Victoria, the analysis is more topical than ever before and we would very much appreciate a rapid review of this new version.

Since the last submission, we have identified an issue with the analysis not raised by the reviewers and outside of our control. Specifically, we have been working through the methods of the papers by Prem et al. (2017 and 2021) that we previously used for the age-stratified synthetic contact matrices that were important inputs to our model. On review of this work, we have formed the opinion that the approach used to creating these synthetic matrices is not satisfactory for producing realistic matrices. In particular, we identified at least 16 significant important errors in the methods used for creating these matrices that could be expected to lead to inaccuracies in the matrices. We attach our analysis of the methods used in this paper to this submission. However, we do not consider these materials to be part of our submission, but rather just provide these as an explanation for why we have changed approach. We have contacted the corresponding author of this paper to consider the implications for modelling more generally, given that this paper is widely cited but appears severely flawed.

Instead of using the synthetic matrices from Prem et al., we have now changed approach to using the original empiric matrices from the POLYMOD study by Mossong et al. with re-weighting to Australia's population distribution, as described in the revised Supplement. This change has led to two knock-on changes to our methodological approach.

First, the pattern of mixing by location changed somewhat with this new approach and required further adjustments to other aspects of our methods. Specifically, because household contacts became somewhat more important than with the Prem matrices, the posterior distribution for the effect of face coverings increased. We believe that this is because a greater effect to reduce the transmission rates in the non-home locations was required to fit the epidemic. The parameter for the effect of face coverings reached a range that we considered unrealistic if the model configuration was left otherwise unchanged, with the posterior estimates under the new approach reaching values of 100% if unconstrained through the prior distributions (and could even have non-zero posteriors at the 100% upper limit, implying values beyond 100% could also have resulted in good fits).

We believe the explanation for this is that our previous approach with the Prem matrices over-emphasised the importance of contacts in the home. In reality, contact saturation is important in residential settings, where transmission to household members could not result in repeat transmission back to the original index case. Further, non-home contacts are more important to perpetuating transmission through linking people with dissimilar contact networks. However, these effects are impossible to capture within the context of a deterministic model. Therefore, to capture these processes, we included a parameter for the relative decrease in importance of home contacts relative to contacts in the three other matrix locations (school, workplace and other). We assigned this parameter an uninformative prior of a 0 to 40% reduction and found it to be unconstrained through the fitting process, but that this new approach recovered a very good calibration fit with plausible values for all other calibration parameters. In particular, the estimate for the effect of face coverings is plausible and consistent with our previous submission.

Second, because the Prem estimates underpin the clinical fraction estimates from Davies et al., we have changed to preferring estimates of this quantity from Sah et al. in PNAS. These values are higher than those we previously used, such that our estimates of the case detection rate have

decreased somewhat. The study by Sah et al. is a high quality analysis that estimates these quantities directly from empiric data without the need for mechanistic model-based inference.

To summarise, since the previous revision, we have changed to what we consider more reliable matrices and added one additional parameter (the relative importance of home contacts relative to the other three matrix locations). We have also responded to the reviewers' comments as follows. In general, although all of the exact values of our results have changed marginally, the conclusions of our analysis are similar to that of the previously submitted version.

1. *"That is, it is not an essential aspect of calibration that all parameters that are varied through the algorithm are constrained." -- This is true, but with the caveat that many of these parameters are constrained from empirical studies in other contexts. The "correct" approach is of course to use these empirical estimates as the prior.*

We agree with the reviewer. We used empirical estimates for parameters that remained fixed throughout and selected a mix of informative and uninformative priors for calibration. These choices were made with consideration of the available evidence for these quantities.

2. *"That is, we believe that the independence of some of the parameters is an interesting and valid finding from the calibration approach and suggests the parameters are constrained by different data sources." -- The way to test this would be to generate surrogate data using AuTuMN using several different parameter sets, add some noise, and then use the algorithm to try to recover the original parameter values -- i.e., test the identifiability of the model. If not too onerous, I would suggest doing this analysis as I still have some concerns about this aspect of the methodology. Note that even fairly simple models can have unidentifiable parameters (<https://www.ncbi.nlm.nih.gov/pmc/articles/PMC7752088/>). For parameters that are determined to be unidentifiable, I appreciate the authors' comment about propagating uncertainty, and here a prior distribution, rather than the posterior, may be as good (or even better) to use.*

We acknowledge that the experiments suggested by the Reviewer would be an appropriate approach for testing the model's identifiability. However, unfortunately the complexity of the model and the relatively large number of calibrated parameters make this type of analysis infeasible for this analysis.

We have now conducted an additional analysis to verify the inference capacity of our model and its calibration (see Section 14 of the Supplement). Namely, we ran a separate calibration where ten parameters were initialised from a more diverse set of starting points in the parameter space using Latin Hypercube Sampling. We demonstrated that convergence still occurred and similar posterior distributions were obtained, compared to the main analysis. This analysis also emphasises the sensitivity of the calibration likelihood to the key parameters estimated in this study (e.g. face coverings effect), demonstrating that these parameters are constrained by the observations used to fit the model.

3. *"However, we would argue that it is even more important for readers to understand the reasons for the forward projections behaving in the way that they do." -- I agree; the comment was more about whether it is really possible to get a more accurate estimate of, e.g., face mask efficacy from calibrating this particular model, rather than using best estimates from the empirical literature.*

Apologies for any misunderstanding.

4. *"It was challenging to encapsulate the factors that made Victoria's experience unique" -- Up to the authors to choose what's best (including leaving as-is), but personally I feel it would be more helpful*

to readers to also acknowledge the similarities between Victoria and other highly successful locations (e.g., Singapore; potentially Taiwan, although they have not yet achieved complete elimination of their most recent outbreak).

We have revised the following text from the last paragraph of the Introduction from -

“Nevertheless, the clear reversal in the trajectory of the epidemic following the implementation of these policy changes offers the opportunity to explore the contribution of these factors to the epidemic profile. Indeed, Victoria’s second wave was virtually unique, in that these policy changes reversed substantial and escalating community cases rates and supported subsequent sustained elimination, which was achieved for several months from November 2020.”

- to -

“Although several countries of Asia maintained effective control strategies through much of 2020,¹³ Victoria’s second wave was notable in that these policy changes reversed substantial and escalating community cases rates and supported subsequent sustained elimination, which was achieved for several months from November 2020. The clear reversal in the trajectory of the epidemic following the implementation of these policy changes offers the opportunity to explore the contribution of these factors to the epidemic profile.”

5. The supplement still uses the terminology "cluster", which I believe has been adapted to "service" in the main text. (Personally I still find these terms confusing if they refer predominantly just to "regions", but I defer to the authors on this point.) I also don't see expanded description of the calibration algorithm (e.g., the 10,000 iterations -- and was a single chain used for each cluster/service?).

Thank you for bringing this to our attention. We have reviewed the Supplement to ensure that the term “cluster” has been replaced with “service” throughout. (Services/clusters are not strictly spatial regions, but historical patterns of hospital presentations from local government areas of Victoria, which makes them more challenging to define.)

The model is implemented as a single simulation of interacting spatial patches (the services) with heterogeneous spatial mixing. Seven calibration chains were run for the model. The calibration approach is described in the Supplement, and we have added one further sentence this description to ensure clarity:

“Seven chains were then run to ensure 10,000 post-burn-in iterations were achieved.”

REVIEWERS' COMMENTS

Reviewer #1 (Remarks to the Author):

The authors are thanked for their revision, the careful attention paid to the contact matrices, and the new analysis in the supplement. In all honesty, I do not find the convergence results convincing, but I do not think we will come to agreement on this point and it is relatively minor compared to the merits of the paper. I do not wish to delay this valuable manuscript any further, and apologize for the delays already caused.